# Characterizing Datapoints via Second-Split Forgetting

**Pratyush Maini**[1]     **Saurabh Garg**[1]     **Zachary C. Lipton**[1]     **J. Zico Kolter**[1,2]
Carnegie Mellon University[1]     Bosch Center for AI[2]
{pratyushmaini,zlipton}@cmu.edu; {sgarg2, zkolter}@cs.cmu.edu

## Abstract

Researchers investigating example hardness have increasingly focused on the dynamics by which neural networks learn and forget examples throughout training. Popular metrics derived from these dynamics include (i) the epoch at which examples are first correctly classified; (ii) the number of times their predictions flip during training; and (iii) whether their prediction flips if they are held out. However, these metrics do not distinguish among examples that are hard for distinct reasons, such as membership in a rare subpopulation, being mislabeled, or belonging to a complex subpopulation. In this paper, we propose *second-split forgetting time* (SSFT), a complementary metric that tracks the epoch (if any) after which an original training example is forgotten as the network is fine-tuned on a randomly held out partition of the data. Across multiple benchmark datasets and modalities, we demonstrate that *mislabeled* examples are forgotten quickly, and seemingly *rare* examples are forgotten comparatively slowly. By contrast, metrics only considering the first split learning dynamics struggle to differentiate the two. At large learning rates, SSFT tends to be robust across architectures, optimizers, and random seeds. From a practical standpoint, the SSFT can (i) help to identify mislabeled samples, the removal of which improves generalization; and (ii) provide insights about failure modes. Through theoretical analysis addressing overparameterized linear models, we provide insights into how the observed phenomena may arise.[1]

## 1  Introduction

A growing literature has investigated metrics for characterizing the difficulty of training examples, driven by such diverse motivations as (i) deriving insights for how to reconcile the ability of deep neural networks to generalize [30] with their ability to memorize noise [15, 48]; (ii) identifying potentially mislabeled examples; and (iii) identifying notably challenging or rare sub-populations of examples. Some of these efforts have turned towards learning dynamics, with researchers noting that neural networks tend to learn cleanly labeled examples before mislabeled examples [17, 18, 33], and more generally tend to learn *simpler* patterns sooner—for several intuitive notions of simplicity [19, 35, 43]. Broadly, works in this area tend to characterize examples as belonging either to *prototypical groups* or *memorized exceptions* [7, 16, 25]. Adapting these intuitions to real datasets, Feldman [15] propose rating the degree to which an example is memorized based on whether its predicted class flips when it is excluded from the training set. These, and other works [8, 21, 35, 43, 47] have proposed many metrics for characterizing example difficulty with Carlini et al. [7] comparing five such metrics. However, while many of these works distinguish some notion of *easy* versus *hard* samples, they seldom (i) offer tools for distinguishing among different types of hard examples; (ii) explain theoretically why these metrics might be useful for distinguishing easy versus hard samples. Moreover, existing metrics tend to give similar scores to examples that are difficult for distinct reasons, e.g, membership in rare, complex, or mislabeled sub-populations.

---

[1]Code for reproducing our experiments can be found at https://github.com/pratyushmaini/ssft.

36th Conference on Neural Information Processing Systems (NeurIPS 2022).

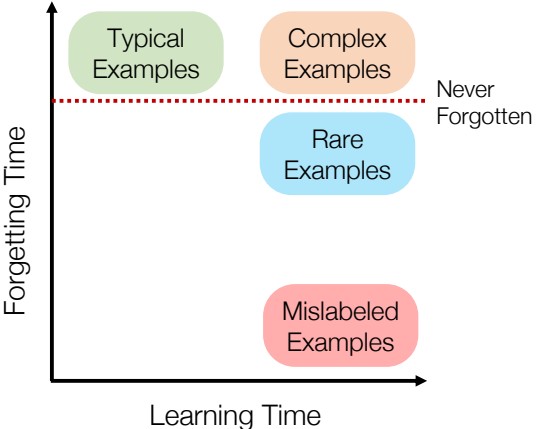

Figure 1: Overview of example separation offered by the unified view of learning and forgetting time.

In this paper, we propose to additionally consider a new metric, Second-Split Forgetting Time (SSFT), calculated based on the forgetting dynamics that arise as training examples are forgotten when a neural network continues to train on a second, randomly held out data partition. SSFT is defined as the fine-tuning epoch after which a first-split training example is no longer classified correctly. We find that SSFT identifies mislabeled examples remarkably well but does little to separate out under- versus over-represented subpopulations. Conversely, metrics based on the (first-split) training dynamics are more discriminative for separating these populations but less useful for detecting mislabeled examples. We leverage the complementarity of first- and second-split metrics, showing that by jointly visualizing the two, we can produce a richer characterization of the training examples.

In our experiments, we operationalize several notions of hard examples, namely: (i) **mislabeled** examples, for which the original label has been flipped to a randomly chosen incorrect label; (ii) **rare** examples, which belong to underrepresented subpopulations; and (iii) **complex** examples, which belong to subpopulations for which the classification task is more difficult (details in Section 3.2). We perform specific ablation studies with datasets complicated by just one type of hard example (Section 4.3), and show how SSFT can help to distinguish among these categories of examples. We observe that during second-split training, neural networks (i) first forget mislabeled examples from the first split; (ii) only slowly begin to forget *rare* examples (e.g., from underrepresented subpopulations) unique to the first training set; and (iii) do not forget complex examples.

This separation of hard example types has multiple practical applications. **First**, we can use the method to identify noisy labels: On CIFAR-10 with 10% added class noise, SSFT achieves 0.94 AUC for identifying mislabeled samples, while the first-split metrics range in AUC between 0.58 to 0.90. **Second**, the method can also help improve generalization in noisy data settings: while the removal of hard examples according to first-split learning time degrades the performance of the classifier, the removal of hard examples according to SSFT can actually *improve* generalization. This is especially beneficial when e.g., training on synthetic data (produced by a generative model) or mislabeled data. **Third**, we show how SSFT can identify failure modes of machine learning models. For example, in a simplified task classifying between horses and airplanes in the CIFAR-10 dataset, we find that training examples containing horses with sky backgrounds and airplanes with green backgrounds are among the earliest forgotten—indicating that the model relies on the background as a spurious feature. **Last**, we also add that our metric is robust across multiple seeds, and the earliest forgotten examples are robust across architectures. Across multiple optimizers, SSFT distinguishes mislabeled samples, whereas first-split metrics appear more sensitive to the choice of optimizer.

Finally, we investigate second-split dynamics theoretically, analyzing overparametrized linear models [46]. We introduce notions of mislabeled, rare, and complex examples appropriate to this toy model. Our analysis shows that mislabeled examples from the first split are forgotten quickly during second-split training whereas rare examples are not. However, as we train for a long time, rare examples from the first split are eventually forgotten as the model converges to the minimum norm solution on the second split while predictions on complex examples remain accurate with high probability.

## 2 Related Work

**Example Hardness.** Several recent works quantify example hardness with various training-time metrics. Many of these metrics are based on first-split learning dynamics [8, 25, 27, 35, 43]. Other works have resorted to properties of deep networks such as compression ability [21] and prediction depth [5]. Carlini et al. [7] study metrics centered around model training such as confidence, ensemble agreement, adversarial robustness, holdout retraining, and accuracy under privacy-preserving training. Closest in spirit to the SSFT studied in our paper are efforts in [7, 47]. Crucially, Carlini et al. [7] study the KL divergence of the prediction vector after fine-tuning on a held-out set at a low learning rate, and do not draw any direct inference of the separation offered by their metric. Focusing on (first-split) forgetting dynamics, Toneva et al. [47] defined a metric based on the number of forgetting events during training and identified sets of *unforgettable* examples that are never misclassified once learned. In our work, we find complementary benefits of analysis based on first- and second-split dynamics.

**Memorization of Data Points.** In order to capture the memorization ability of deep networks, their ability to memorize noise (or randomly labeled samples) has been studied in recent work [3, 48]. As opposed to the memorization of rare examples, the memorization of noisy samples hurts generalization and makes the classifier boundary more complex [15]. On the contrary, a recent line of works has argued how memorization of (atypical) data points is important for achieving optimal generalization performance when data is sampled from long-tailed distributions [6, 11, 15].

**Simplicity Bias.** Another line of work argues that neural networks have a bias toward learning simple features [43], and often do not learn complex features even when the complex feature is more predictive of the true label than the simple features. This suggests that models end up memorizing (through noise) the few samples in the dataset that contain the complex feature alone, and utilize the simple feature for correctly predicting the other training examples [1, 32].

**Label Noise.** Large-scale machine learning datasets are typically labeled with the help of human labelers [12] to facilitate supervised learning. It has been shown that a significant fraction of these labels are erroneous in common machine learning datasets [39]. Learning under noisy labels is a long-studied problem [2, 26, 31, 37]. Various recent methods have also attempted to identify label noise [10, 23, 38, 40]. While the focus of our work is not to propose a new method in this long line of work, we show that the view of forgetting time naturally distills out examples with noisy labels. Future work may benefit by augmenting our metric with SOTA methods for label noise identification.

## 3 Method

The primary goal of our work is to *characterize* the hardness of different datapoints in a given dataset. Suppose we have a dataset $\mathcal{S}_A = \{\mathbf{x}_i, \mathbf{y}_i\}^n$ such that $(\mathbf{x}_i, \mathbf{y}_i) \sim \mathcal{D}$. For the purpose of characterization, we augment each datapoint $(\mathbf{x}_i, \mathbf{y}_i) \in \mathcal{S}_A$ with parameters $(\mathbf{fslt}_i, \mathbf{ssft}_i)$ where $\mathbf{fslt}_i$ quantifies the first-split learning time (FSLT), and $\mathbf{ssft}_i$ quantifies the second-split forgetting time (SSFT) of the sample. To obtain these parameters, we next describe our proposed procedure.

**Procedure** We train a model $f$ on $\mathcal{S}$ to minimize the empirical risk: $\mathcal{L}(\mathcal{S}; f) = \sum_i \ell(f(\mathbf{x}_i), \mathbf{y}_i)$. We use $f_A$ to denote a model $f$ (initialized with random weights) trained on $\mathcal{S}_A$ until convergence (100% accuracy on $\mathcal{S}_A$). We then train a model initialized with $f_A$ on a held-out split $\mathcal{S}_B \sim \mathcal{D}^n$ until convergence. We denote this model with $f_{A \to B}$. To obtain parameters $(\mathbf{fslt}_i, \mathbf{ssft}_i)$, we track per-example predictions $(\hat{\mathbf{y}}_i^t)$ at the end of every epoch ($t^{\text{th}}$) of training. Unless specified otherwise, we train the model with cross-entropy loss using Stochastic Gradient Descent (SGD).

**Definition 1** (First-Split Learning Time). *For $\{\mathbf{x}_i, \mathbf{y}_i\} \in \mathcal{S}_A$, learning time is defined as the earliest epoch during the training of a classifier $f$ on $\mathcal{S}_A$ after which it is always classified correctly, i.e.,*

$$\mathbf{fslt}_i = \operatorname*{argmin}_{t^*} (\hat{\mathbf{y}}_{i,(A)}^t = \mathbf{y}_i \ \ \forall t \geq t^*) \quad \forall \{\mathbf{x}_i, \mathbf{y}_i\} \in \mathcal{S}_A. \tag{1}$$

**Definition 2** (Second-Split Forgetting Time). *Let $\hat{\mathbf{y}}_{i,(A \to B)}^t$ to denote the prediction of sample $\{\mathbf{x}_i, \mathbf{y}_i\} \in \mathcal{S}_A$ after training $f_{(A \to B)}$ for $t$ epochs on $\mathcal{S}_B$. Then, for $\{\mathbf{x}_i, \mathbf{y}_i\} \in \mathcal{S}_A$ forgetting time is defined as the earliest epoch after which it is never classified correctly, i.e.,*

$$\mathbf{ssft}_i = \operatorname*{argmin}_{t^*} (\hat{\mathbf{y}}_{i,(A \to B)}^t \neq \mathbf{y}_i \quad \forall t \geq t^*) \quad \forall \{\mathbf{x}_i, \mathbf{y}_i\} \in \mathcal{S}_A. \tag{2}$$

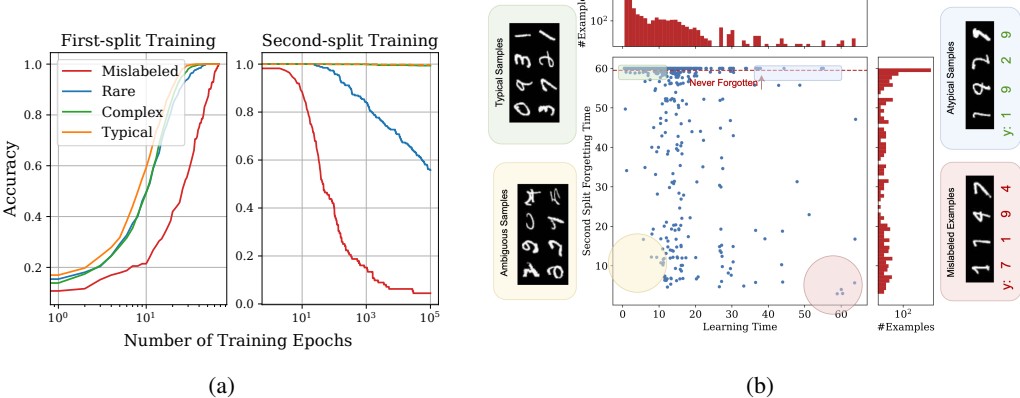

(a)          (b)

Figure 2: Rate of Learning and Forgetting of examples for different groups in the synthetic dataset. While first-split training is not able to distinguish between rare and complex examples, second-split training succeeds in distinguishing them. Additionally, second-split training separates mislabeled examples from the rest relatively better than first-split training. (b) Visualization of first-split learning and second-split forgetting times when training LeNet model on the MNIST dataset.

## 3.1 Baseline Methods

We provide a brief description of metrics for example hardness considered in recent comparisons [25].

**Number of Forgetting Events:** ($n_f$). An example $(\mathbf{x}_i, \mathbf{y}_i) \in \mathcal{S}$ undergoes a forgetting event when the accuracy on the example decreases between two consecutive updates. Toneva et al. [47] analyzed the total number of such events $n_f$ during the training of a neural network to identify hard examples.

**Cumulative Learning Accuracy:** ($\mathrm{acc}_l$). Jiang et al. [25] suggest that rather than using the learning time (Definition 1), using the number of epochs during training when a machine learning model correctly classifies a given sample is a more stable metric for predicting example hardness.

**Cumulative Learning Confidence:** ($\mathrm{conf}_l$). Similar to $\mathrm{acc}_l$, $\mathrm{conf}_l$ measures the cumulative softmax confidence of the model towards the correct class over the course of training.

## 3.2 Example Characterization

We characterize example hardness via three sources of learning difficulty: **(i) Mislabeled Examples:** We refer to mislabeled examples as those datapoints whose label has been flipped to an incorrect label uniformly at random. **(ii) Rare Examples:** We assume that rare examples belong to sub-populations of the original distribution that have a low probability of occurrence. In particular, there exist $O(1)$ examples from such sub-populations in a given dataset. In Section 4.3 we describe how we operationalize this notion in the case of the CIFAR-100 dataset. **(iii) Complex Examples:** These constitute samples that are drawn from sub-groups in the dataset that require either (1) a hypothesis class of high complexity; or (2) higher sample complexity to be learnt relative to examples from rest of the dataset. We leave the definition of complex samples mathematically imprecise, but with the same intuitive sense as in prior work [3, 43]. For instance, in a dataset composed of the union of MNIST and CIFAR-10 images, we would consider the subpopulation of CIFAR-10 images to be more *complex*.

## 4 Empirical Investigation of First- and Second-Split Training Dynamics

### 4.1 Experimental Setup

**Datasets** We show results on a variety of image classification datasets—MNIST [13], CIFAR-10 [29], and Imagenette [22]. For experiments in the language domain, we use the SST-2 dataset [45]. For each of the datasets, we split the training set into two equal partitions $(\mathcal{S}_A, \mathcal{S}_B)$. For experiments

| Sentences in SST-2 dataset with smallest forgetting time | Label |
|---|---|
| The director explores all three sides of his story with a sensitivity and an inquisitiveness reminiscent of Truffaut | Neg |
| Beneath the film's obvious determination to shock at any cost lies considerable skill and determination , backed by sheer nerve | Neg |
| This is a fragmented film, once a good idea that was followed by the bad idea to turn it into a movie | Pos |
| The holiday message of the 37-minute Santa vs. the Snowman leaves a lot to be desired. | Pos |
| Epps has neither the charisma nor the natural affability that has made Tucker a star | Pos |
| The bottom line is the piece works brilliantly | Neg |
| Alternative medicine obviously has its merits ... but Ayurveda does the field no favors | Pos |
| What could have easily become a cold, calculated exercise in postmodern pastiche winds up a powerful and deeply moving example of melodramatic moviemaking | Neg |
| Lacks depth | Pos |
| Certain to be distasteful to children and adults alike , Eight Crazy Nights is a total misfire | Pos |

Table 1: First-split sentences that were forgotten by the 3rd epoch of second-split training of a BERT-base model on the SST-2 dataset. Notice that all of these forgotten examples are mislabeled.

with mislabeled examples, we simulate mislabeled examples by randomly selecting a subset of 10% examples from both the partitions and changing their label to an incorrect class.

**Training Details**  Unless otherwise specified, we train a ResNet-9 model [4] using SGD optimizer with weight decay 5e-4 and momentum 0.9. We use the cyclic learning rate schedule [44] with a peak learning rate of 0.1 at the 10th epoch. We train for a maximum of 100 epochs or until we have 5 epochs of 100% training accuracy. We first train on $\mathcal{S}_A$, and then using the pre-initialized weights from stage 1, train on $\mathcal{S}_B$ with the same learning parameters. All experiments can be performed on a single RTX2080 Ti. Complete hyperparameter details are available in Appendix B.1.

## 4.2   Learning-Forgetting Spectrum for various datasets

**Synthetic Dataset**  We consider data $(\mathbf{x}, \mathbf{y})$ sampled from a mixture of multiple distributions $\mathcal{D}_g$, s.t. $\mathbf{x} \in \mathbb{R}^d$. $\mathcal{D}_g$ denotes the $g^{\text{th}}$ group and has a sampling frequency of $\pi_g$. Each group $\mathcal{D}_g \equiv (\mathcal{X}_g, \{\mathbf{y}_g\})$, i.e., the true label for all the samples drawn from a given group is the same, and the examples in each group are non-overlapping. Each group is parametrized by a set of $k \ll d$ unique indices $\mathcal{I}_g \subset [d]$ such that $\mathcal{I}_i \cap \mathcal{I}_j = \phi$ for $i \neq j$. The discriminative characteristic of each group is the vector $\mathbf{u}_g$, such that, $[\mathbf{u}_g]_i = 1$ if $i \in \mathcal{I}_g$ else $0 \ \forall i \in [d]$. Then for any sample $(\mathbf{x}, \mathbf{y}) \in \mathcal{S}$:

$$P(\mathbf{x} \in \mathcal{X}_g) = \pi_g; \quad \mathbf{x}|\mathcal{X}_g \sim \mathcal{N}(0, \sigma^2 \mathbf{I}_d) + \boldsymbol{\mu}_g.$$

For our simulation, we consider a 10 class-classification problem, with $\mu_g = 5$ for typical groups, and $\mu_g = 4$ for complex groups (higher signal to noise ratio). For any sample drawn from a rare group, we have $O(1)$ samples from that group in the entire dataset ($\mathcal{S}_A \cup \mathcal{S}_B$). Mislabeled samples are only generated from the majority typical groups. In Figure 2a, we show the rate of learning and forgetting of examples from each of these categories. We note that in the second-split training, the mislabeled examples are quickly forgotten, and the complex examples are never forgotten. The rare examples are forgotten slowly. In Section 5 we will theoretically justify the observations in the synthetic dataset and show that the rare examples are expected to be forgotten as we train for an infinite time.

**Image Domain**  In Figure 2b, we show representative examples in the four quadrants of the learning-forgetting spectrum. More specifically, we find that the examples forgotten fastest and learned last are mislabeled. And the ones learned early and never forgotten once learned are characteristic simple examples of the MNIST dataset. Examples in the first and third quadrant are seemingly atypical and ambiguous respectively. Similar visualizations for other image datasets can be found in Appendix B.2.

**Other Modalities**  The forgetting and learning dynamics occur broadly across modalities apart from images. We repeat the same problem setup on the SST-2 [45] dataset for sentiment classification. We fine-tune a pre-trained BERT-base model [14] successively on two disjoint splits of the dataset. In Table 1, we provide a list of the earliest forgotten samples when we train a BERT model on the second split of SST-2 dataset. The results suggest that SSFT is able to identify mislabeled samples.

## 4.3   Ablation Experiments

We design specific experimental setups to capture the three notions of hardness as defined in Section 3.

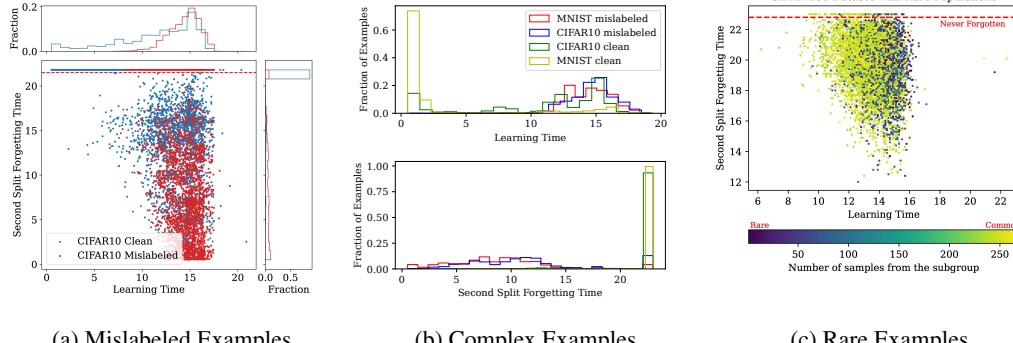

(a) Mislabeled Examples       (b) Complex Examples       (c) Rare Examples

Figure 3: Ablation experiments to distinguish the learning and forgetting dynamics for specific types of hard examples. (a) Mislabeled samples may be learned as slowly as a high fraction of typical samples, but they are forgotten much faster. (b) FSLT distinguishes complex (CIFAR10 clean) and simple (MNIST clean) examples, but SSFT does not. On the contrary, FSLT can not distinguish clean and mislabeled examples of CIFAR10, while the SSFT can. (c) FSLT is able to distinguish examples based on the sub-group frequency, however, SSFT has a low correlation with the sub-group frequency.

**Mislabeled Examples** We sample 10% datapoints from both the first and second split of the CIFAR-10 dataset, and randomly change their label to an incorrect label. Figure 3a shows the learning-forgetting spectrum for the dataset. In the adjoining density histograms, note that a large fraction of the mislabeled and correctly labeled examples are learned at the same time. However, during second-split training, the mislabeled examples are forgotten quickly whereas a large fraction of the clean examples are never forgotten, allowing SSFT to succeed in distinguishing mislabeled samples.

**Complex Examples** We generate a joint dataset that contains the union of both MNIST and CIFAR-10 examples. This is motivated by work in simplicity bias [43] that argue that neural networks learn simpler features first. We also add 10% labeled noise to each of the datasets in the union to understand the learning and forgetting time relationship of a sample that is complex or mislabeled together. In Figure 3b, we show the FSLT and SSFT for MNIST and CIFAR-10 samples. We note that a high fraction of the CIFAR-10 (complex) samples learn at the same speed as the mislabeled samples. However, when looking at the SSFT, we are able to draw a strong separation between the mislabeled samples and complex samples. This indicates that the complexity of a sample has low correlation with its tendency to be forgotten once learnt, but a high correlation with being learned slowly.

**Rare Examples** The CIFAR-100 [29] dataset is a 100-class classification task. The dataset contains 20 superclasses, each containing 5 subclasses. We create a 20-class classification dataset with long tails simulated through the 5 sub-classes within each superclass. More specifically, the number of examples in each subgroup for a given superclass is given by {500, 250, 125, 64, 32} respectively (exponentially decaying with a factor of 2). This is done to simulate the hypothesis of dataset subgroups following a Zipf distribution [49] as argued for by Feldman [15]. This dataset is further divided into two equal splits to analyze the learning-forgetting dynamics. In order to remove any other effects of example hardness (either within a subgroup, or among subgroups), we randomize both the chosen subset of examples and the ordering of the majority and minority groups between each superclass, by training the model on 20 such random splits and aggregating learning and forgetting statistics over these runs. In Figure 3c, we show a scatter plot for the FSLT and SSFT, colored by the frequency of the group a particular example belongs to. We observe that FSLT strongly correlates with the size of the subgroup, whereas the SSFT has a very low correlation with the rareness of a sample.

We provide further ablations to show that FSLT is able to identify hard and rare examples, but SSFT shows nearly no discriminative power at finding the two in Appendix C.

## 4.4 Dataset Cleansing

**Identifying Label Noise** We present AUC scores for detection of label noise via various popular methods in example difficulty literature, across various datasets in Table 2. We note that (i) cumulative predictions over the course of training help stabilize both the learning time and forgetting time metrics;

| Method → | fslt | $\mathbf{acc}_l$ | ssft (Ours) | $\mathbf{acc}_f$ (Ours) | $\mathbf{conf}_l$ | $\mathbf{n}_f$ | Joint (Ours) |
|---|---|---|---|---|---|---|---|
| Imagenette | 0.834 | 0.912 | 0.931 | 0.941 | 0.786 | 0.781 | **0.957** |
| CIFAR10 | 0.740 | 0.900 | 0.938 | 0.941 | 0.947 | 0.580 | **0.958** |
| MNIST | 0.973 | **0.998** | 0.997 | **0.998** | 0.965 | 0.377 | **0.998** |
| CIFAR100 | 0.700 | 0.899 | 0.865 | 0.885 | 0.860 | 0.300 | **0.926** |
| EMNIST | 0.987 | 0.990 | 0.987 | 0.989 | 0.984 | 0.386 | **0.997** |

Table 2: AUC for identification of label noise using various metrics for example hardness across different datasets. Across all datasets, our **ssft** metric outperforms alternative baselines. We introduce $\mathbf{acc}_f$ as the cumulative accuracy on the second-split training, inspired by previous work that suggests using cumulative accuracies helps make first-split learning time more stable [25]. All other notations are described in Section 3. In the case of the Joint method, we select new prediction ranks based on the combined learning and forgetting ranks, further improving over the **ssft** metric alone.

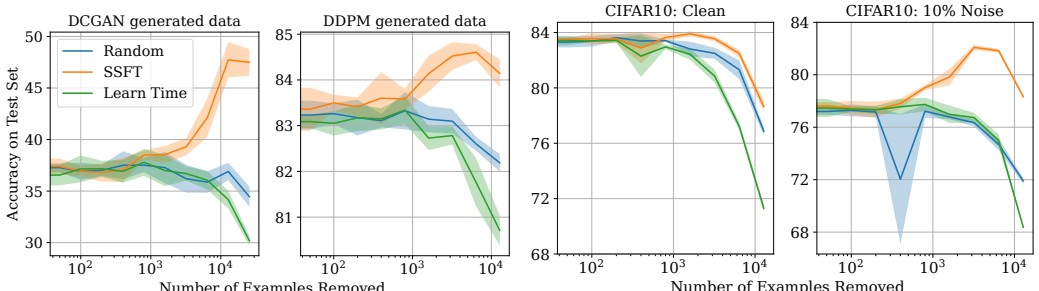

(a) Synthetically generated CIFAR10-like data          (b) CIFAR10 with and without label noise

Figure 4: Accuracy on CIFAR-10 test set after removing a varying number of samples from the training set based on (i) random choice, (ii) examples with the lowest SSFT, and (ii) the highest FSLT. Removing examples based on SSFT helps improve the generalization on the original test set.

(ii) for simple datasets such as MNIST with few ambiguous images, all of the baseline methods have very high AUC (greater than 0.99) in finding noisy inputs. However, in datasets such as CIFAR-10 and Imagenette, we find that second-split forgetting metrics do better than first-split training metrics. Finally, we also compare the use of both forgetting and learning time to find noisy samples, and we find a small improvement in the results of just using the forgetting time. While we do not make explicit comparisons with other state of art methods dedicated to finding label noise, our results suggest that augmenting second split forgetting time information may help improve their results. As also observed in recent work [25], we find that the number of forgetting events ($\mathbf{n}_f$) [47] is an unreliable indicator of mislabeled samples. We hypothesize that this is because of the fact that mislabeled examples may often be (first) learnt very late, hence their count of total forgetting events is also low.

**Cleaning synthetically generated datasets**     Generative models are capable of mimicking the distribution of a given dataset. We generate synthetic datasets of CIFAR10-like samples using (i) DDPM (denoising diffusion model [20]); and (ii) DCGAN (Deep Convolutional GAN [41]). In both cases, we assign pseudo-labels using the BiT model [28] as in prior work [36]. We collect a sample of 50,000 training examples and record the generalization performance on CIFAR-10 as we remove 'hard' samples, as evaluated by various metrics. In Figure 4, we can see that removing the most easily forgotten examples can benefit by up to 10% generalization accuracy on the clean test set of CIFAR-10. In case of the synthetic data generated using DDPM, the gains in generalization performance are under 2%. We hypothesize that this is because the samples generated by DDPM are more representative of the typical distribution of CIFAR-10 than those generated by DCGAN.

**Note:** The ability to train on a second split allows SSFT the *unique* opportunity to train on a clean split of CIFAR-10 in order to assess the alignment of the synthetic samples with the oracle samples. As a result, the SSFT is much more effective in filtering out ambiguous first-split synthetic examples.

## 4.5 Evaluating Example Utility

Recent works [16, 47] have argued for removing a large fraction of the less memorized examples, and keeping the memorized ones. We will analyze the change in model generalization upon removing varying sizes of examples from the training set, as ranked by lowest SSFT and highest FSLT (Figure 4). In the presence of noisy examples, removing samples based on the SSFT helps improve generalization, whereas FSLT does not do much better than random. We draw the following inferences:

**FSLT finds important samples**   As we remove more samples from the dataset, the accuracy of the model trained after samples are removed based on the highest FSLT is significantly lower than random guessing. This suggests that the utility of these samples is higher than random samples. Put in line with the hypothesis of memorization of rare example as proposed in [15], we see that empirically, the examples that are slow to learn are important for the model's test set generalization.

**SSFT removes pathological samples**   On the contrary, removing examples based on the SSFT helps improve model generalization (especially when there is label noise). Even in the setting when there is no label noise, in contrast to FSLT, we find that removing examples that were easily forgotten has a lower negative impact on the model's generalization as opposed to removing random samples. This suggests that the examples that are forgotten in the early epochs of second-split training hurt a model's generalization, and may not be characteristic samples of their particular class.

**Practitioner's view**   From the AUC numbers in Table 2, it may appear that removing examples via learning-based metrics such as learning time and cumulative learning accuracy also provides a high rate of removal of noisy samples. However, when we observe the example utility graphs in Figure 4, we draw the inference that the examples that are learned late, are often important examples (such as rare memorized examples). However, even when SSFT fails to capture the correct noisy examples, it still removes unimportant samples and does not hurt generalization. Similar graphs for other metrics described in Table 2 can be found in Appendix B.

## 4.6 Characterizing Potential Failure Modes

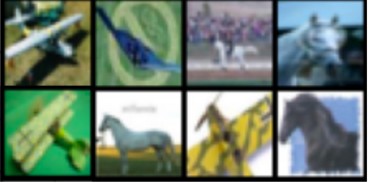

Recent works have attempted to train classifiers on datasets that contain spurious features [24, 42] (example Waterbirds, CelebA [34] dataset). However, a fundamental challenge is to first identify the spurious correlation that the classifier may be relying on. Only then can recent methods be trained to remove the reliance on spurious patterns. We train a ResNet-9 model to classify CIFAR-10 images of horses and airplanes. In Figure 5, we observe that the model forgets planes with green backgrounds and horses with blue backgrounds. This suggests that the model relied on the background as a spurious feature. By analyzing the forgotten examples we can further investigate the examples that the classifier fails to generalize to.

Figure 5: By inspecting the earliest-forgotten examples, we can gain insights into potential failure modes. This model quickly forgets planes with green backgrounds and horses on blue backgrounds.

**Stability of SSFT**   We note that SSFT is stable across multiple seeds (Pearson correlation of 0.81), and across architectures (Pearson correlation of 0.63). While the overall correlation for samples ranked by SSFT may be low across architectures, the top-ranked examples have a high correlation (0.85), suggesting the most forgotten examples are consistent across architectures. In contrast, FSLT has a Pearson correlation of 0.52 across seeds. Most interestingly, the learning time metric is brittle to the choice of hyperparameters. As shown by Jiang et al. [25], when using Adam optimizer, examples of different hardness get learned together. In our experiments, we observe the same phenomenon during learning, however, SSFT is robust to the choice of the optimizer. Detailed results in Appendix C.1.

**Limitations**   One limitation of the proposed metric is that it is brittle to the choice of the learning rate for the second-split training. If we use a very small learning rate, then overparametrized deep models are capable of learning the new dataset without forgetting examples from the first split. Alternately, if we use a very large learning rate, the model may diverge and undergo catastrophic forgetting. However, under 'reasonable' choices of learning rate (like that for first-split training), we find SSFT is robust. We provide a detailed anaylsis of the same in Appendix C.1.

# 5 Theoretical Results

Through our theoretical analysis, we will characterize the forgetting dynamics of mislabeled, rare and complex examples in a simplified version of the framework used for our synthetic experiments in Figure 2a. Recall, our setup contains two dataset splits $\mathcal{S}_A, \mathcal{S}_B$, where we train on the first split until achieving perfect accuracy on all training points, and then with these weights train on $\mathcal{S}_B$ for infinite time. In particular, we will prove that both mislabeled and rare examples are forgotten upon training for infinite time, with mislabeled examples being forgotten much faster. Further, we will show that complex examples from the first split do not get forgotten if not continually trained on. We assume in our analysis that $\mathcal{S}_B$ has no mislabeled or rare examples, and $\mathcal{S}_A$ contains one example of each type.

We consider a dataset $\mathcal{S} = \{\mathbf{x}_i, \mathbf{y}_i\}^n$ such that $(\mathbf{x}_i, \mathbf{y}_i) \in \mathcal{X} \times \mathcal{Y}$, and $\mathbf{x}_i = \boldsymbol{\mu}_g + \mathbf{z}_i$ where $\mathbf{z}_i \sim \mathcal{N}\left(0, \sigma^2 \mathbf{I}_d\right)$, and $\|\boldsymbol{\mu}_g\|_2^2 = k\mu^2$ (as in Section 4.2). Let $\mathbf{w} \in \mathbb{R}^d$ represent the weight vector of an overparametrized linear model. We analyze the learning and forgetting dynamics by minimizing the empirical risk: $\mathcal{L}(\mathcal{S}; \mathbf{w}) = \sum_i \ell(\mathbf{w}^\top \mathbf{y}_i \mathbf{x}_i)$, where $\ell$ is the exponential loss. Following Chatterji and Long [9], we make the following assumptions about the problem setup:

**(A.1)** The failure probability satisfies $0 \leq \delta \leq 1/C$,

**(A.2)** The number of samples satisfies $n \geq C \log(1/\delta)$,

**(A.3)** The input dimension $d \geq C \max\{n^2 \log(n/\delta), n(k \cdot \mu^2/\sigma^2)\}$, and $k \cdot \mu^2/\sigma^2 \geq C \log(n/\delta)$,

where $k \cdot \mu^2/\sigma^2$ represents the signal to noise ratio and $C$ is a large constant. Now we formalize the notions of rare, mislabeled and complex examples for our theoretical analysis.

**Definition 1** (Rare Examples, $\mathcal{R}$ [15])**.** *Consider a dataset $\mathcal{S}$ sampled from a mixture of distributions $\{\mathcal{D}_1, \ldots, \mathcal{D}_N\}$ with frequency $\{\pi_1, \ldots, \pi_N\}$ respectively. Let $\mathcal{R} \subseteq \mathcal{S}$ be the set of rare examples. Then, for all $(\mathbf{x}_i, \mathbf{y}_i) \in \mathcal{R}$, if $(\mathbf{x}_i, \mathbf{y}_i) \sim \mathcal{D}_j$, then there are $O(1)$ samples from $\mathcal{D}_j$ in $\mathcal{S}$.*

**Definition 2** (Mislabeled Examples, $\mathcal{M}$)**.** *Consider a $k$ class classification problem with $\mathcal{Y} = \{1, 2, \ldots, k\}$. Let $\mathcal{M} \subset \mathcal{S}$ be the set of mislabeled examples. Then for any $(\mathbf{x}, \mathbf{y}) \sim \mathcal{D}$, a corresponding mislabeled example is given by $(\mathbf{x}, \tilde{\mathbf{y}}) \in \mathcal{M}$ such that $\tilde{\mathbf{y}} \in \mathcal{Y} \setminus \{\mathbf{y}\}$.[2]*

**Definition 3** (Complex Examples, $\mathcal{C}$)**.** *Let $\mathcal{C} \subset \mathcal{S}$ be the set of examples sampled from complex distributions. Let $(\mathbf{x}_i, \mathbf{y}_i) \in \mathcal{C}$ such that $(\mathbf{x}_i, \mathbf{y}_i) \sim \mathcal{D}_g$ (complex group), then $\mu_g = \frac{\mu_t}{\lambda}$, $\lambda > 1$ where $\mu_t$ is the coordinate-wise mean for samples drawn from any simple distribution $\mathcal{D}_t$ (Section 4.2).*

**Optimization** We perform gradient descent with fixed learning rate $\eta$,

$$\mathbf{w}(t+1) = \mathbf{w}(t) - \eta \nabla \mathcal{L}(\mathbf{w}(t)) = \mathbf{w}(t) - \eta \sum_i \ell'(\mathbf{w}^\top \mathbf{y}_i \mathbf{x}_i) \cdot \mathbf{y}_i \mathbf{x}_i. \tag{3}$$

**Solution dynamics** For sufficiently small learning rate $\eta$, and (bounded) starting point $\mathbf{w}(0)$, Soudry et al. [46] showed that:

$$\mathbf{w}(t) = \hat{\mathbf{w}} \log t + \rho(t), \tag{4}$$

where $\rho(t)$ is a bounded residual term, and $\hat{\mathbf{w}}$ is the solution to the hard margin SVM:

$$\hat{\mathbf{w}} = \underset{\mathbf{w} \in \mathbb{R}^d}{\operatorname{argmin}} \ \|\mathbf{w}\|_2^2 \quad s.t. \ \mathbf{w}^\top \mathbf{y}_i \mathbf{x}_i \geq 1, \tag{5}$$

## 5.1 First-split Learning

For stage 1, we consider that we train the model for a maximum of $T$ epochs (until we achieve 100% accuracy on the first training dataset $\mathcal{S}_A$). This means that the learned weight vectors are close to, but have not converged to the max margin solution. The solution at the end of $t$ epochs is given by $\mathbf{w}_A(t)$. At sufficiently large $T$, we have:

$$\mathbf{w}_A(T) = \hat{\mathbf{w}}_A \log T + \rho_A(T)$$
$$\mathbf{w}_A(T)^\top \mathbf{y}_i \mathbf{x}_i \geq 1 \quad \forall (\mathbf{x}_i, \mathbf{y}_i) \in \mathcal{S}_A \tag{6}$$

---

[2]For binary classification, $\mathcal{Y} = \{-1, +1\}$. The labels are reversed for mislabeled examples.

## 5.2 Second-split Forgetting

We initialize the weights for second stage of training with $\mathbf{w}_A(T)$ from first training stage, and then train on $\mathcal{S}_B$. We provide the formal theorem statement and complete proofs in Appendix A, but provide informal theorem statements and an intuitive proof sketch below:

**Theorem 1** (Asymptotic Forgetting (informal)). *For sufficiently small learning rate, given datasets $\mathcal{S}_A, \mathcal{S}_B \sim \mathcal{D}^n$. After training for $T' \to \infty$ epochs, the following hold with high probability:*

1. *Mislabeled and Rare examples from $\mathcal{S}_A$ are forgotten.*
2. *Complex examples from $\mathcal{S}_A$ are not forgotten.*

*Proof Sketch.* We use the result from Soudry et al. [46] that for any bounded initialization, when trained on a separable data, the model converges to the same min-norm solution. As a result, we can ignore the impact of $\mathcal{S}_A$ at infinite time training. Then, we use generalization bounds from Chatterji and Long [9] to argue about the accuracy on mislabeled and complex examples. For the case of rare examples, we show that the probability of correct model prediction can be approximated by a Gaussian CDF with mean 0 and $\mathcal{O}(1/\sqrt{n})$ variance.

**Theorem 2** (Intermediate-Time Forgetting (informal)). *For sufficiently small learning rate, given two datasets $\mathcal{S}_A, \mathcal{S}_B \sim \mathcal{D}^n$. For a model initialized with weights, $\mathbf{w}_B(0) = \mathbf{w}_A(T)$ and trained for $T' = f(T)$ epochs, the following hold with high probability:*

1. *Mislabeled examples from $\mathcal{S}_A$ are no longer incorrectly predicted.*
2. *Rare examples from $\mathcal{S}_A$ are not forgotten.*

*Proof Sketch.* $\mathcal{S}_B$ contains examples from the same majority distributions as $\mathcal{S}_A$. The mislabeled example also belongs to one of these distributions, but has the opposite label. However, $\mathcal{S}_B$ does not have samples from rare groups found in $\mathcal{S}_A$. Using representer theorem, we decompose the model updates into a weighted sum of each training data point in $\mathcal{S}_B$. Then, we analyze the change in prediction on rare and mislabeled examples, which is a dot product of the weight update with $\mathbf{x}_m$ or $\mathbf{x}_r$. Per our assumptions, the the mean of each group $\boldsymbol{\mu}_g$ is orthogonal to the other. As a result, the rare example finds negligible coupling with any example in $\mathcal{S}_B$, and the variance of its prediction keeps increasing due to the noise term contributed in the model weights from each example in $\mathcal{S}_B$. On the contrary, the mislabeled examples have a strong coupling with all the examples in its group. Due to its incorrect label, the mean of its predictions moves towards the correct label, with variance increasing at a similar rate. The final step is to jointly analyze the rate of change of prediction of both the examples, and find an optimal time $T'$ when the prediction on the mislabeled example is flipped and the rare example still retains its prediction with high probability.

## 6 Conclusion

While many prior works investigate training time dynamics to characterize the hardness of examples, we enrich this literature with a complementary lens focused on the second-split forgetting time. We demonstrate the potential of SSFT to distinguish among rare, mislabeled, and complex examples; and also show the differences in the example properties captured by first-split and second-split metrics.

Our work opens new lines of inquiry in future work that can utilize the separation of hard examples. First, we expect state of art methods in label noise identification to benefit by augmenting our approach. Further, we believe our ablations showing that complex, noisy, and mislabeled samples may all be learned slowly inspire future work that can unite different takes on the memorization-generalization research—early learning, simplicity bias, and singleton memorization.

## Acknowledgements

We would like to thank Aakash Lahoti and Jeremy Cohen for their insightful comments on this work. SG acknowledges Amazon Graduate Fellowship and JP Morgan PhD Fellowship for their support. ZL acknowledges Amazon AI, Salesforce Research, Facebook, UPMC, Abridge, the PwC Center, the Block Center, the Center for Machine Learning and Health, and the CMU Software Engineering Institute (SEI) via Department of Defense contract FA8702-15-D-0002, for their generous support.

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
