# Characterizing Datapoints via Second-Split Forgetting
## Supplementary Material

## A  Theoretical Results

### A.1  Preliminaries

Let $\mathbf{w} \in \mathbb{R}^d$ represent the weight vector of overparametrized linear model. We analyze the learning and forgetting dynamics by minimizing the empirical risk: $\mathcal{L}(\mathcal{S}; \mathbf{w}) = \sum_i \ell(\mathbf{w}^\top \mathbf{y}_i \mathbf{x}_i)$. We consider the exponential loss $\ell(z) = \exp(-z)$ for our analysis. For completeness, we rewrite the definitions and preliminaries from the main paper below.

**Data Generating Process**  We restate the data generating process as detailed in the synthetic experiment in Section 4.2. We consider data $(\mathbf{x}, \mathbf{y})$ sampled from a mixture of multiple distributions $\mathcal{D}_g$, s.t. $\mathbf{x} \in \mathbb{R}^d$. $\mathcal{D}_g$ denotes the $g^{\text{th}}$ group and has a sampling frequency of $\pi_g$. Each group $\mathcal{D}_g$ is a distribution over $(\mathcal{X}_g \times \{\mathbf{y}_g\})$, i.e., the true label for all the samples drawn from a given group is the same, and the examples in each group are non-overlapping. Each group is parametrized by a set of $k \ll d$ unique indices $\mathcal{I}_g \subset [d]$ such that $\mathcal{I}_i \cap \mathcal{I}_j = \phi$ for $i \neq j$. The discriminative characteristic of each group is the vector $\mathbf{u}_g$, such that, $[\mathbf{u}_g]_i = 1$ if $i \in \mathcal{I}_g$ else $0 \; \forall i \in [d]$. In the following discussion, we will refer to $\mu_g$ as the coordinate-wise mean vector for the group $g$, such that $\boldsymbol{\mu}_g = \mu_g \mathbf{u}_g$. Then for any sample $(\mathbf{x}, \mathbf{y}) \in \mathcal{S}$:

$$P(\mathbf{x} \in \mathcal{X}_g) = \pi_g; \quad \mathbf{x}|\mathcal{X}_g \sim \mathcal{N}(0, \sigma^2 \mathbf{I}_d) + \boldsymbol{\mu}_g. \tag{7}$$

**Definition 1** (Rare Examples, $\mathcal{R}$ [15]). *Consider a dataset $\mathcal{S}$ sampled from a mixture of distributions $\{\mathcal{D}_1, \ldots, \mathcal{D}_N\}$ with frequency $\{\pi_1, \ldots, \pi_N\}$ respectively. Let $\mathcal{R} \subseteq \mathcal{S}$ be the set of rare examples. Then, for all $(\mathbf{x}_i, \mathbf{y}_i) \in \mathcal{R}$, if $(\mathbf{x}_i, \mathbf{y}_i) \sim \mathcal{D}_j$, then there are $O(1)$ samples from $\mathcal{D}_j$ in $\mathcal{S}$.*

**Definition 2** (Mislabeled Examples, $\mathcal{M}$). *Consider a $k$ class classification problem with $\mathcal{Y} = \{1, 2, \ldots, k\}$. Let $\mathcal{M} \subset \mathcal{S}$ be the set of mislabeled examples. Then for any $(\mathbf{x}, \mathbf{y}) \sim \mathcal{D}$, a corresponding mislabeled example is given by $(\mathbf{x}, \tilde{\mathbf{y}}) \in \mathcal{M}$ such that $\tilde{\mathbf{y}} \in \mathcal{Y} \setminus \{\mathbf{y}\}$.*[1]

**Definition 3** (Complex Examples, $\mathcal{C}$). *Let $\mathcal{C} \subset \mathcal{S}$ be the set of examples sampled from complex distributions. Let $(\mathbf{x}_i, \mathbf{y}_i) \in \mathcal{C}$ such that $(\mathbf{x}_i, \mathbf{y}_i) \sim \mathcal{D}_g$ (complex group), then $\mu_g = \frac{\mu_t}{\lambda}$, $\lambda > 1$ where $\mu_t$ is the coordinate-wise mean for samples drawn from any simple distribution $\mathcal{D}_t$ (Section 4.2).*

The implication of the aforementioned characterization is that complex distributions have a lower signal-to-noise ratio (SNR) as compared to simple distributions. We assume the sample complexity required to estimate the distribution as a proxy for the complexity of the distribution. In this regard, having a low SNR increases the complexity.

Recall that $\mathcal{S}_A$ and $\mathcal{S}_B$ denote the first and second training splits of our dataset. In our theoretical framework, we only consider two majority distributions in the second split dataset ($\mathcal{S}_B$). Let us call them $\mathcal{D}_1, \mathcal{D}_2$. Therefore, both $\mathcal{S}_A$ and $\mathcal{S}_B$ contain $O(n)$ samples from $\mathcal{D}_1, \mathcal{D}_2$. In the first split dataset ($\mathcal{S}_A$), we consider the presence of another distribution $\mathcal{D}_r$ that constitutes the rare example (only one sample $(\mathbf{x}_r, \mathbf{y}_r)$). The mislabeled example (only one sample $(\mathbf{x}_m, \mathbf{y}_m)$) belongs to one of the majority distributions, and we will assume without loss of generality that this is from distribution $\mathcal{D}_1$. To understand the population accuracy in the case of complex distributions, we will draw a simple analogy in the case when the majority distributions $\mathcal{D}_1, \mathcal{D}_2$ occur from complex distributions as defined below. Based on equation 7, without loss of generality, we will assume that dimensions $\{1 \ldots k\}, \{k + 1 \ldots 2k\}, \{2k + 1 \ldots 3k\}$ are the predictive dimensions for the majority group 1, 2 and that for the rare example from dataset 1. We make these assumptions to simplify the theoretical exposition. However, our results can be observed even after relaxing them at the expense of more book-keeping.

Based on Chatterji and Long [9], we make the following assumptions about the problem setup:

**(A.1)**  The failure probability satisfies $0 \leq \delta \leq 1/C$,

**(A.2)**  The number of samples satisfies $n \geq C \log(1/\delta)$,

---

[1] For binary classification, $\mathcal{Y} = \{-1, +1\}$. The labels are reversed for mislabeled examples.

**(A.3)** The input dimension $d \geq C \max\{n^2 \log(n/\delta), nk\mu^2/\sigma^2\}$, and $k \cdot \mu^2/\sigma^2 \geq C \log(n/\delta)$, where $k \cdot \mu^2/\sigma^2$ represents the signal to noise ratio.

**Optimization**   We perform gradient descent with fixed learning rate $\eta$,

$$\mathbf{w}^{(t+1)} = \mathbf{w}(t) - \eta \nabla \mathcal{L}(\mathbf{w}(t)) = \mathbf{w}(t) - \eta \sum_i \ell'(\mathbf{w}^\top \mathbf{y}_i \mathbf{x}_i) \cdot \mathbf{y}_i \mathbf{x}_i. \tag{8}$$

**Asymptotic Solution** [46]   For sufficiently small learning rate $\eta$, and (bounded) starting point $\mathbf{w}(0)$,

$$\mathbf{w}(t) = \hat{\mathbf{w}} \log t + \rho(t), \tag{9}$$

where $\hat{\mathbf{w}}$ is the solution to the hard margin SVM:

$$\hat{\mathbf{w}} = \underset{\mathbf{w} \in \mathbb{R}^d}{\operatorname{argmin}} \ \|\mathbf{w}\|_2^2 \ \ s.t. \ \mathbf{w}^\top \mathbf{y}_i \mathbf{x}_i \geq 1, \tag{10}$$

## A.2   Learning Stage

For the stage 1, we consider that we train the model for a maximum of $T$ epochs (until we achieve 100% accuracy on the first training dataset $\mathcal{S}_A$ with margin greater than 1). This means that the learned weight vectors are close to, but have not converged to the max margin solution. The solution at the end of $T$ epochs is given by $\mathbf{w}_A(t)$. At sufficiently large $T$, we have:

$$\begin{aligned} \mathbf{w}_A(T) &= \hat{\mathbf{w}}_A \log T + \rho_A(T) \\ \mathbf{w}_A(T)^\top \mathbf{y}_i \mathbf{x}_i &\geq 1 \quad \forall (\mathbf{x}_i, \mathbf{y}_i) \in \mathcal{S}_A \end{aligned} \tag{11}$$

**Lemma 1** (Bounded Weights). *With probability at least* $1 - \delta$*, there exists a bounded time t beyond which the model classifies all training points correctly.*

*Proof.* From Lemma 5, we know that the dataset is separable with probability at least $1 - \delta$. This means that the max-margin solution (or the SVM solution) for the dataset classifies all training points correctly. From the analysis in Soudry et al. [46], we know that:

$$\begin{aligned} \lim_{t \to \infty} \frac{\mathbf{w}_A(t)}{\|\mathbf{w}_A(t)\|_2} &= \frac{\hat{\mathbf{w}}_A}{\|\hat{\mathbf{w}}_A\|_2}, \\ \lim_{t \to \infty} \mathbf{y}_i \mathbf{x}_i^\top \frac{\mathbf{w}_A(t)}{\|\mathbf{w}_A(t)\|_2} &= \mathbf{y}_i \mathbf{x}_i^\top \frac{\hat{\mathbf{w}}_A}{\|\hat{\mathbf{w}}_A\|_2} \end{aligned} \tag{12}$$

Then it directly follows that, for $\epsilon > 0$, there exists bounded time $T > 0$ such that $\forall t > T$,

$$\left| \mathbf{y}_i \mathbf{x}_i^\top \frac{\mathbf{w}_A(t)}{\|\mathbf{w}_A(t)\|_2} - \mathbf{y}_i \mathbf{x}_i^\top \frac{\hat{\mathbf{w}}_A}{\|\hat{\mathbf{w}}_A\|_2} \right| < \epsilon$$

This concludes that there exists a time T for which the sign of the prediction of both the max-margin solution and the learnt solution will be the same, implying correct prediction for every example in the training set for a bounded time solution.

$\square$

## A.3   Forgetting Stage

We initialize the weights for second stage of training with $\mathbf{w}_A(T)$ from first training stage, and then train on $\mathcal{S}_B$ to minimize the empirical loss using gradient descent (Equation 3). Assume that the dataset is balanced in the class labels, $|\mathcal{S}_B| = n$. Also, recall that $\mathcal{S}_B$ does not contain mislabeled or rare examples from the same sub-group as in $\mathcal{S}_A$. For analyzing example forgetting in an asymptotic setting, we will directly borrow results from the analysis made by Chatterji and Long [9]. They prove a stronger result for the case where the dataset contains a fraction $\eta$ of mislabeled examples. However, we will use the setting without label noise. The asymptotic result then builds on to the main theorem of the paper on intermediate time forgetting (Theorem 4).

**Transformations for Equivalence to Chatterji and Long [9]** In our setup, we consider a group structure where each distribution has a mean vector that is orthogonal to the others. We show the equivalence of the same to the data model studied in prior work [9].

Let us define $\boldsymbol{\mu}_1, \boldsymbol{\mu}_2 \in \mathbb{R}^d$ as follows:

$$[\boldsymbol{\mu}_1]_j = \begin{cases} \mu & \text{if } j \in \{1 \ldots k\} \\ 0 & \text{o.w.} \end{cases}$$

Similarly, define $\boldsymbol{\mu}_2, \boldsymbol{\mu}_3$ as well. For the equivalence condition, we are only concerned about the dataset split $\mathcal{S}_B$ which is where the generalization bounds hold. Further, let $z \sim \mathcal{N}\left(0, \sigma^2 \mathbf{I}_d\right)$. Then, $\mathbf{x}|\mathcal{X}_g \sim z + \boldsymbol{\mu}_g$.

We can now shift and rescale the axes such that the new origin is located at $(\boldsymbol{\mu}_1 + \boldsymbol{\mu}_2)/2$. Then, define $\boldsymbol{\mu} = (\boldsymbol{\mu}_1 - \boldsymbol{\mu}_2)/2$. This results in the simplification that the mean of the examples sampled from $\mathcal{D}_1$ is $\boldsymbol{\mu}$ and that from $\mathcal{D}_2$ is $-\boldsymbol{\mu}$. We can hence express $\mathbf{x} = \mathbf{y}\boldsymbol{\mu} + \mathbf{z}$. This directly follows their model assumption where $\mathbf{z} \in \mathbb{R}^d$ has each marginal sampled from a mean zero subgaussian distribution with subgaussian norm at most 1. In our case, each marginal is a Gaussian random variable with variance $\sigma^2$. Once again, we can rescale the axes such that $\tilde{\boldsymbol{\mu}} = \boldsymbol{\mu}/\sigma$. Now, our data model directly follows the data model discussed in prior work [9].

## A.4 Asymptotic Analysis

First, we analyze infinite-time training case. We will extend the result from Chatterji and Long [9], Soudry et al. [46] to show that (i) mislabeled and rare examples are forgotten when the model is trained for long; and (ii) complex examples are not forgotten. First, recall the result used in Subsection A.2 for any bounded initialization for weights $\mathbf{w}_B(0)$,

$$\mathbf{w}_B(t) = \hat{\mathbf{w}}_B \log t + \rho_B(t). \tag{13}$$

We will first provide a formal version of Theorem 1 which was informally stated in the main paper.

**Theorem 3** (Asymptotic Forgetting). *Under assumptions A.1, A.2, A.3, with probability 1-$\delta$, fine-tuning for $t \to \infty$ iterations on the second dataset $\mathcal{S}_B$ produces a max-margin classifier $\hat{\mathbf{w}}_B$ such that*

$$\mathbb{P}_{(\mathbf{x}_m, \mathbf{y}_m) \in \mathcal{S}_A} \left[\text{sign}(\hat{\mathbf{w}}_B \cdot \mathbf{x}_m) = \mathbf{y}_m\right] \leq \exp\left(-c\|\tilde{\boldsymbol{\mu}}\|_2^2/d\right) ,$$

$$\Phi\left(-1/C\right) \leq \mathbb{P}_{(\mathbf{x}_r, \mathbf{y}_r) \in \mathcal{S}_A} \left[\text{sign}(\hat{\mathbf{w}}_B \cdot \mathbf{x}_r) \neq \mathbf{y}_r\right] \leq \Phi\left(1/C\right) ,$$

$$\mathbb{P}_{(\mathbf{x}_c, \mathbf{y}_c) \in \mathcal{S}_A} \left[\text{sign}(\hat{\mathbf{w}}_B \cdot \mathbf{x}_c) \neq \mathbf{y}_c\right] \leq \exp\left(-c\|\tilde{\boldsymbol{\mu}}/\lambda\|_2^2/d\right) ,$$

*for some absolute constant c>0 and $(\mathbf{x}_m, \mathbf{y}_m) \in \mathcal{M}, (\mathbf{x}_r, \mathbf{y}_r) \in \mathcal{R}, (\mathbf{x}_c, \mathbf{y}_c) \in \mathcal{C}$ respectively.*

The theorem implies that the probability that mislabeled examples from $\mathcal{S}_A$ are classified with the given (incorrect) label tends to 0 if $\|\boldsymbol{\mu}\|_2 = \theta(d^\beta)$ for any $\beta \in (1/4, 1/2]$. Note that in our case, we have $k$ dimensions of signal. Therefore, as long as $k/d$ is a constant fraction, as the input dimensions $d \to \infty$ the above holds. Examples in $\mathcal{S}_A$ from distributions absent in $\mathcal{S}_B$ are randomly classified.

*Intuition.* In the asymptotic case, the initial weights from first split training $\mathbf{w}_A(T) = \mathcal{O}(\log T) \ll \mathbf{w}_B(t)$, where $t \to \infty$ in the limit of infinite training. As a consequence, for any bounded initialization, the model weights converge to the minimum norm solution (SVM) solution from Soudry et al. [46]. We use results from [9] who study the setting of a binary classification problem with noisy label fraction $\eta$. In our case, since the second-split is assumed to have only clean samples, $\eta = 0$. The final step is to adapt our data generating process to the format used in Chatterji and Long [9].

*Proof.* Recall the transformations described in Section A.3. Let $(\mathbf{x}_m, \mathbf{y}_m), (\mathbf{x}_r, \mathbf{y}_r), (\mathbf{x}_c, \mathbf{y}_c)$ represent any point from $\mathcal{S}_A$ which belongs to mislabeled set $\mathcal{M}$, rare set $\mathcal{R}$ and complex set $\mathcal{C}$. The important thing to note in the analysis that follows is that each of these examples is independent of the samples in $\mathcal{S}_B$. Hence, the probability of correctly predicting on them is same as that of correctly predicting on a population sample, in the limit of infinite training (when initialization does not matter and all models converge to the same solution).

$$\mathbb{P}_{(\mathbf{x}, \mathbf{y}) \sim \mathcal{D}} \left[\text{sign}(\mathbf{w} \cdot \mathbf{x}) \neq \mathbf{y}\right] = \mathbb{P}_{(\mathbf{x}, \mathbf{y}) \sim \mathcal{D}} \left[(\mathbf{y}\mathbf{w} \cdot \mathbf{x}) < 0\right] ,$$

We will now analyze the probability of correct prediction of mislabeled, rare, and complex examples separately.

**Mislabeled Examples** Since $\mathcal{S}_B$ is separable, in the limit of infinite-training time, the classifier correctly predicts all the examples in the dataset. We will use this fact to show that it assigns the opposite label to the mislabeled sample in set $\mathcal{S}_A$ with high probability. Then, we denote 'failure' as the event that the mislabeled samples is still predicted with label $\mathbf{y}_m$ at infinite-time training.

$$
\begin{aligned}
\mathbb{P}_{(\mathbf{x},\mathbf{y})\sim\mathcal{D}_1}\left[(\mathbf{y}\mathbf{w}\cdot\mathbf{x}) < 0\right] &= 1 - \mathbb{P}\left[(\mathbf{y}_m\mathbf{w}\cdot\mathbf{x}_m) < 0\right] \\
&= \mathbb{P}\left[(\mathbf{y}_m\mathbf{w}\cdot\mathbf{x}_m) > 0\right] \\
&= \mathbb{P}\left[\mathrm{sign}(\mathbf{w}\cdot\mathbf{x}_m) = \mathbf{y}_m\right],
\end{aligned}
$$

Then, we can directly borrow the result from Chatterji and Long [9] (Theorem 4) to find that

$$
\begin{aligned}
\mathbb{P}_{(\mathbf{x},\mathbf{y})\sim\mathcal{D}_1}\left[(\mathbf{y}\mathbf{w}\cdot\mathbf{x}) > 0\right] &\leq \exp\left(-c\frac{||\tilde{\boldsymbol{\mu}}||^4}{d}\right) \\
\mathbb{P}\left[\mathrm{sign}(\mathbf{w}\cdot\mathbf{x}_m) = \mathbf{y}_m\right] &\leq \exp\left(-c\frac{||\tilde{\boldsymbol{\mu}}||^4}{d}\right),
\end{aligned}
\tag{14}
$$

**Rare Examples** Without loss of generality, we may assume that the correct label for the rare example $\mathbf{y}_r = 1$.

$$
\begin{aligned}
\mathbb{P}\left[(\mathbf{y}_r\mathbf{w}\cdot\mathbf{x}_r) < 0\right] &= \mathbb{P}\left[(\mathbf{y}_r\mathbf{w}\cdot(\mathbf{x}_r - \boldsymbol{\mu}_r)) < -\mathbf{y}_r\mathbf{w}\cdot\boldsymbol{\mu}_r\right] \\
&= \mathbb{P}\left[(\mathbf{w}\cdot(\mathbf{x}_r - \boldsymbol{\mu}_r)) < -\mathbf{w}\cdot\boldsymbol{\mu}_r\right], \qquad (\text{since } \mathbf{y}_r = 1) \\
&= \mathbb{P}\left[(\mathbf{w}\cdot\mathbf{z}_r < -\mathbf{w}\cdot\boldsymbol{\mu}_r\right] \\
&= \Phi\left(\frac{-\mathbf{w}\cdot\boldsymbol{\mu}_r}{\sigma\|\mathbf{w}\|_2}\right),
\end{aligned}
\tag{15}
$$

where $\Phi$ is the Gaussian CDF. In the last step we use the fact that $\mathbf{z}_r \sim \mathcal{N}(0, \sigma^2\mathbf{I}_d)$. Therefore its dot product with the vector $\mathbf{w}$ results with a summation of $d$ Gaussian vectors, each with mean 0 and variance $\sigma^2[\mathbf{w}]_i^2 \ \forall i \in [d]$. Now what remains is to prove that the value at which we want to calculate the CDF is close to 0, so that the probability of the misclassification is close to 0.5.

Recall from the analysis in Soudry et al. [46] that asymptotically the model converges to the max-margin separator that is comprised of a weighted sum of the support vectors of the dataset (let us call this set $\mathcal{V}_B$). This means that the final weights of the model $\mathbf{w}$ is a combination of datapoints from the first two majority distributions $\mathcal{D}_1, \mathcal{D}_2$.

$$
\mathbf{w} = \sum_{i\in\mathcal{V}_B} \alpha_i\mathbf{x}_i,
\tag{16}
$$

$$
\boldsymbol{\mu}_r \cdot \mathbf{w} = \sum_{i\in\mathcal{V}_B} \alpha_i(\boldsymbol{\mu}_r \cdot \mathbf{x}_i)
\tag{17}
$$

Since $\boldsymbol{\mu}_r \cdot \boldsymbol{\mu}_i = 0$, we have that $(\boldsymbol{\mu}_r \cdot \mathbf{x}_i) = \boldsymbol{\mu}_r \cdot \mathbf{z}_i + 0$, which is a mean zero random variable.

$$
\boldsymbol{\mu}_r \cdot \mathbf{w} = \sum_{i\in\mathcal{V}_B} \alpha_i(\boldsymbol{\mu}_r \cdot \mathbf{z}_i),
\tag{18}
$$

$$
= \sum_{i\in\mathcal{V}_B} \alpha_i\mu \sum_{2k+1\leq j\leq 3k} [\mathbf{z}_i]_j,
\tag{19}
$$

$$
= \xi,
\tag{20}
$$

$$
\tag{21}
$$

where $\xi \sim \mathcal{N}(0, k\mu^2\sigma^2\sum_{i\in\mathcal{V}_B}\alpha_i^2)$. Also, $\|\mathbf{w}\|_2^2 = \sum_{i\in\mathcal{V}_B}\alpha_i^2\mathbf{x}_i^2$. However, $\mathbf{x}_i = \boldsymbol{\mu}_i + \mathbf{z}_i$. From Lemma 4, we know that with probability greater than $1-\delta/6$, for every example, $\frac{d\sigma^2}{2} \leq \|\mathbf{z}_i\|_2^2 \leq \frac{3d\sigma^2}{2}$. By Young's inequality for products:

$$\|\mathbf{z}_i\|_2^2 = \|\mathbf{x}_i - \boldsymbol{\mu}_i\|_2^2,$$
$$\leq 2\|\mathbf{x}_i\|_2^2 + 2\|\boldsymbol{\mu}_2\|_2^2,$$
$$\|\mathbf{x}_i\|_2^2 \geq \frac{1}{2}\|\mathbf{z}_i\|_2^2 - \|\boldsymbol{\mu}_i\|_2^2,$$
$$\geq \frac{d\sigma^2}{4} - \|\boldsymbol{\mu}_i\|_2^2,$$
$$\geq \frac{d\sigma^2}{8} \qquad \text{(since } \frac{k\mu^2}{\sigma^2} < d/nC \text{ for large C)},$$
$$\|\mathbf{w}\|_2^2 = \sum_{i\in\mathcal{V}_B} \alpha_i^2\mathbf{x}_i^2 \geq \frac{d\sigma^2}{8}\sum_{i\in\mathcal{V}_B} \alpha_i^2.$$

Therefore,

$$\eta = \frac{-\xi}{\sigma\|\mathbf{w}\|_2} \sim \mathcal{N}\left(0, \frac{k\mu^2}{d\sigma^2/8}\right),$$
$$\eta \sim \mathcal{N}\left(0, \frac{1}{nC}\right), \qquad \text{(since } \frac{k\mu^2}{\sigma^2} < d/nC \text{ for large C)}, \tag{22}$$
$$\mathbb{P}\left[(\mathbf{y}_r\mathbf{w}\cdot\mathbf{x}_r) < 0\right] = \Phi\left(\eta\right)$$

Using Gaussian tail bound on $\eta$, with probability at least $1 - \delta$, $\eta \leq \sqrt{\frac{\log(1/\delta)}{nC}}$. Therefore,

$$\mathbb{P}\left[(\mathbf{y}_r\mathbf{w}\cdot\mathbf{x}_r) < 0\right] \leq \Phi\left(\sqrt{\frac{\log(1/\delta)}{nC}}\right) \leq \Phi\left(\frac{1}{C}\right) \approx 0.5.$$

In the last step, we used the assumption (A.2) that $n \geq C\log(1/\delta)$ for some large constant C.

*Remark:* This analysis highlights that the meaning of being 'forgotten' for rare examples is to predict randomly. However, in case of mislabeled examples, the predicted label of the example approaches its 'true' label, irrespective of its training label.

**Complex Examples** The analysis for complex examples follows directly from the analysis in the case of mislabeled examples. The only difference is the magnitude of the signal to noise ratio within the group. Recall that complex examples are also sampled from majority groups. Therefore, the probability that the complex example $(\mathbf{x}, \mathbf{y})$ is misclassified at the end of training for infinite time is given by:

$$\mathbb{P}\left[\mathbf{y}\mathbf{x}^\top\mathbf{w} < 0\right] \leq \exp\left(-c\frac{\|\bar{\boldsymbol{\mu}}/\lambda\|^4}{d}\right) \tag{23}$$

This approaches 0, indicating perfect classification of complex examples from $\mathcal{S}_A$. Note that this is a complimentary case where the second split only has examples from the complex distribution.

$\square$

## A.5 Intermediate Time Analysis

From the analysis in Section A.4, we find that all the mislabeled and rare examples are forgotten by the time we train for $t \to \infty$ iterations. Since examples from complex subgroups are not forgotten even at infinite time training, we skip analysis for those examples in this subsection. Our goal is to show that there exists a time $T'$ such that with high probability, the model forgets all the mislabeled examples, but still correctly classifies all the rare examples. To show this, we will track the accuracy of inputs in the first data split, as we train on the examples in the second split.

The model output for any sample $(\mathbf{x}_i, \mathbf{y}_i) \in \mathcal{S}_A$ is given by $\mathbf{w}(t)^\top\mathbf{x}_i$, and the prediction is considered to be correct if $\text{sign}(\mathbf{w}(t)^\top\mathbf{x}_i) = \mathbf{y}_i$, or if $\mathbf{w}(t)^\top\mathbf{x}_i\mathbf{y}_i > 0$. From hereon, we will use the notation $\mathbf{a}_i^t = \mathbf{w}(t)^\top\mathbf{x}_i\mathbf{y}_i$.

Recall that we considered that there is one example from both the mislabeled and rare example category in $\mathcal{S}_A$. We will denote these data points by $(\mathbf{x}_m, \mathbf{y}_m) \in \mathcal{M}$, and $(\mathbf{x}_r, \mathbf{y}_r) \in \mathcal{R}$ respectively. Without loss of generality, we will assume that $(\mathbf{x}_m, \neg \mathbf{y}_m)$ was sampled from $\mathcal{D}_1$ in the mixture of distributions $\mathcal{D}$. All the examples in the second split $\mathcal{S}_B$ sampled from the same distribution are given by $\mathcal{S}_{B,1} \subset \mathcal{S}_B$. The remaining examples are in the set denoted by $\mathcal{S}_{B,\neg 1} \subset \mathcal{S}_B$.

From the learning time training dynamics, we know that the initialization of the weights for the second round of training is given by:

$$\mathbf{w}_B(0) = \hat{\mathbf{w}}_A \log T + \rho_A(T). \tag{24}$$

Now, from the representer theorem, we can decompose the model weights at any iteration of second-split training as follows,

$$\mathbf{w}_B(t) = \hat{\mathbf{w}}_A \log T + \rho_A(T) + \sum_{j \in \mathcal{S}_B} \beta_j \mathbf{y}_j \mathbf{x}_j. \tag{25}$$

Note that we introduce an additional term $\mathbf{y}_j$ in the decomposition using representer theorem, but this can be done without loss of generality since $\mathbf{y}_j \in \{-1, +1\}$. This helps us in ensuring that each $\beta_j$ is non-negative as shown in Lemma 6.

From Lemma 8, we know that there exists a bounded time $T'$ when the mislabeled example prediction flips. We will denote $\mu$ as the coordinate-wise mean for the $k$-signal dimensions in the vector $\boldsymbol{\mu}_g$ in the discussion that follows. Let us define $\Delta_t = \frac{\sum_{j \in \mathcal{S}_{B,1}} \beta_j(t)}{\sum_{j \in \mathcal{S}_B} \beta_j(t)}$, and $\Delta = \max_t \Delta_t$. For a symmetric distribution with two majority subgroups with the opposite label, we would expect this value to be close to 0.5. Now, we present formal version of Theorem 2 from the main paper.

**Theorem 4** (Intermediate-Time Forgetting). *Under the distribution outlined in Appendix A.1 with assumptions A.1, A.2, A.3, whenever $\mathcal{S}_A$ is separable, when fine-tuning on the second dataset split $\mathcal{S}_B$ with sufficiently small learning rate, there exists some bounded time T' when*

1. $\mathbb{P}_{(\mathbf{x}_m, \mathbf{y}_m) \in \mathcal{S}_A}[\mathbf{y}_m \neq \mathbf{w}(T') \cdot \mathbf{x}_m] \geq 1 - c_0 \exp\left(\frac{-k^2 \mu^2 \Delta^2}{c \sigma^2 d}\right) - c_1 \exp(-cd)$

2. $\mathbb{P}_{(\mathbf{x}_r, \mathbf{y}_r) \in \mathcal{S}_A}[\mathbf{y}_r = \mathbf{w}(T') \cdot \mathbf{x}_r] \geq 1 - c_0 \exp\left(\frac{-k^2 \mu^2 \Delta^2}{c \sigma^2 d}\right) - c_1 \exp(-cd)$

*for absolute constants $c_0, c_1, c$ and $(\mathbf{x}_m, \mathbf{y}_m) \in \mathcal{M}, (\mathbf{x}_r, \mathbf{y}_r) \in \mathcal{R}$ respectively.*

If the fraction of dimensions that contain the signal, $k/d$, is considered fixed then both the first and second term above exponentially decay with a factor of $d$. This suggests that increasing overparametrization leads to a higher likelihood of the phenomenon of intermediate time forgetting—there exists an epoch when the prediction of mislabeled example is flipped but the rare examples are still correctly predicted with high probability.

In what follows, the key idea is to show that at a time when the prediction of the mislabeled example is incorrect with high probability, the predictions of the rare examples is correct with high probability.

*Proof.* From Lemma 7, we know that $(\mathbf{x}_m, \mathbf{y}_m)$ and $(\mathbf{x}_r, \mathbf{y}_r)$ are support vectors for $\mathcal{S}_A$. Hence, $\mathbf{y}_m \mathbf{x}_m^\top \hat{\mathbf{w}}_A = \mathbf{y}_r \mathbf{x}_r^\top \hat{\mathbf{w}}_A = 1$. We will use this fact to find the distribution for $\mathbf{a}_m^t$, $\mathbf{a}_r^t$.

Let $[\mathbf{x}_j]_i$ denote the $i^{th}$ dimension of the $j^{th}$ datapoint in the second split. Recall that the second split comprises of two majority subgroups. $k$ dimensions of the input vector contain the true signal for class prediction. The dimensions $\{1 \ldots k\}, \{k+1 \ldots 2k\}, \{2k+1 \ldots 3k\}$ are the predictive dimensions for the majority group 1, 2 and that for the rare example from dataset $\mathcal{S}_A$. Therefore, $[\mathbf{x}_j]_i = \mu + [\mathbf{z}_j]_i$, if $i$ is in the predictive dimensions, otherwise $[\mathbf{x}_j]_i = [\mathbf{z}_j]_i$ where $[\mathbf{z}_j]_i \sim \mathcal{N}(0, \sigma^2)$. To make notation simple, we will refer to $\beta_j(t)$ by using $\beta_j$.

*Remark:* We do not perform input transformation to prove the following results.

**Mislabeled Example**    The prediction on the mislabeled point times the given label can be written as:

$$\mathbf{a}_m^t = \mathbf{y}_m \mathbf{x}_m^\top \mathbf{w}_B(t) = \mathbf{y}_m \mathbf{x}_m^\top \left( \hat{\mathbf{w}}_A \log T + \rho_A(T) + \sum_{j \in \mathcal{S}_B} \beta_j \mathbf{y}_j \mathbf{x}_j \right),$$

$$= \log T + c_m + \sum_{j \in \mathcal{S}_B} \beta_j (\mathbf{y}_m \mathbf{x}_m^\top \mathbf{y}_j \mathbf{x}_j) \qquad \text{(using Lemma 7)},$$

(26)

where $c_m = \mathbf{y}_m \mathbf{x}_m^T \rho_A(T)$ is a residual term that does not continue grow with $T$ (Theorem 9 [46]).

Without loss of generality, assume that the mislabeled example $\mathbf{x}_m$ belongs to majority group 1 (true label = 1), and was originally labeled such that $\mathbf{y}_m$ = -1 in the first-split dataset.

$$\sum_{j \in \mathcal{S}_B} \beta_j (\mathbf{y}_m \mathbf{x}_m^\top \mathbf{y}_j \mathbf{x}_j) = \sum_{j \in \mathcal{S}_{B,1}} \beta_j (\mathbf{y}_m \mathbf{x}_m^\top \mathbf{y}_j \mathbf{x}_j) + \sum_{j \in \mathcal{S}_{B,\neg 1}} \beta_j (\mathbf{y}_m \mathbf{x}_m^\top \mathbf{y}_j \mathbf{x}_j)$$

$$= - \sum_{j \in \mathcal{S}_{B,1}} \beta_j (\mathbf{x}_m^\top \mathbf{x}_j) + \sum_{j \in \mathcal{S}_{B,\neg 1}} \beta_j (\mathbf{x}_m^\top \mathbf{x}_j).$$

(27)

Now we can use the distribution properties of the dataset to further simplify per dimension and aggregate the sum across all examples.

$$\sum_{j \in \mathcal{S}_{B,1}} \beta_j (\mathbf{x}_m^\top \mathbf{x}_j) = \mathbf{x}_m^\top \sum_{j \in \mathcal{S}_{B,1}} \beta_j \mathbf{x}_j = \mathbf{x}_m^\top \mathbf{x}_{\mathcal{S}_{B,1}},$$

(28)

where $[\mathbf{x}_{\mathcal{S}_{B,1}}]_i = \mu \, \mathbb{1}(i \in \{1 \ldots k\}) \sum_{\mathcal{S}_{B,1}} \beta_j + [\mathbf{z}_{\mathcal{S}_{B,1}}]_i$, where $[\mathbf{z}_{\mathcal{S}_{B,1}}]_i \sim \mathcal{N}(0, \sigma^2 \sum_{j \in \mathcal{S}_{B,1}} \beta_j^2)$. Now, we will add up the dot product dimension wise. Let us call $B_1 = \sum_{\mathcal{S}_{B,1}} \beta_j$ and $B_1^v = \sum_{j \in \mathcal{S}_{B,1}} \beta_j^2$. Then $[\mathbf{x}_{\mathcal{S}_{B,1}}]_i \sim \mathcal{N}(\mu \, \mathbb{1}(i \leq k)) \cdot B_1, \sigma^2 B_1^v)$.

$$\mathbf{x}_m^\top \mathbf{x}_{\mathcal{S}_{B,1}} = \mu \sum_k [\mathbf{z}_{\mathcal{S}_{B,1}}]_i + \mu B_1 \sum_k [\mathbf{z}_m]_i + k \cdot \mu^2 \cdot B_1 + \sum_d [\mathbf{z}_{\mathcal{S}_{B,1}}]_i \cdot [\mathbf{z}_m]_i$$

$$= \mu \cdot \alpha_1 + \sum_d [\mathbf{z}_{\mathcal{S}_{B,1}}]_i \cdot [\mathbf{z}_m]_i$$

$$\mathbf{x}_m^\top \mathbf{x}_{\mathcal{S}_{B,\neg 1}} = \mu \cdot \alpha_2 + \sum_d [\mathbf{z}_{\mathcal{S}_{B,\neg 1}}]_i \cdot [\mathbf{z}_m]_i$$

(29)

where $\alpha_1 \sim \mathcal{N}(k \cdot \mu \cdot B_1, k \cdot \sigma^2 (B_1^v + B_1^2))$ and $\alpha_2 \sim \mathcal{N}(0, k \cdot \sigma^2 (B_2^v + B_2^2))$. Notice that in the first step, the first three terms are independent of each other and can be added directly to obtain a new random variable using the independence condition, however, the last term is dependent on the first two. In the final step, we also add the terms for the dot product corresponding to the opposite group.

This gives us the overall expression as follows:

$$\sum_{j \in \mathcal{S}_B} \beta_j (\mathbf{y}_m \mathbf{x}_m^\top \mathbf{y}_j \mathbf{x}_j) = \mu \cdot (\alpha_2 - \alpha_1) + \sum_d ([\mathbf{z}_{\mathcal{S}_{B,\neg 1}}]_i - [\mathbf{z}_{\mathcal{S}_{B,1}}]_i) \cdot [\mathbf{z}_m]_i,$$

$$= \mu \cdot \alpha_m + \sum_d [\mathbf{z}_{\mathcal{S}_B}]_i \cdot [\mathbf{z}_m]_i - k \cdot \mu^2 \cdot B_1,$$

(30)

where $\alpha_m \sim \mathcal{N}(0, k \cdot \sigma^2 (B^v + B_1^2 + B_2^2))$ and $[\mathbf{z}_{\mathcal{S}_B}]_i \sim \mathcal{N}(0, \sigma^2 B^v)$, since $(B_1^v + B_2^v = \sum_{j \in \mathcal{S}_B} \beta_j^2 = B^v)$.

$$\mathbf{a}_m^t = \log T + c_m + \xi_m - k \cdot \mu^2 \cdot B_1; \text{ s.t. } \xi_m = \mu \cdot \alpha_m + \sum_d [\mathbf{z}_{\mathcal{S}_B}]_i \cdot [\mathbf{z}_m]_j.$$

(31)

**Rare Example**  Following the analysis of the mislabeled example, we can similarly find an expression for $\mathbf{a}_r^t$ on the rare example as follows:

$$
\begin{aligned}
\mathbf{a}_r^t = \mathbf{y}_r \mathbf{x}_r^\top \mathbf{w}_B(t) &= \mathbf{y}_r \mathbf{x}_r^\top (\hat{\mathbf{w}}_A \log T + \rho_A(T) + \sum_{j \in \mathcal{S}_B} \beta_j \mathbf{y}_j \mathbf{x}_j), \\
&= \log T + c_r + \sum_{j \in \mathcal{S}_B} \beta_j (\mathbf{y}_r \mathbf{x}_r^\top \mathbf{y}_j \mathbf{x}_j) \qquad \text{(using Lemma 7)},
\end{aligned}
\tag{32}
$$

Following the same procedure as for mislabeled example, we get the overall expression as follows:

$$
\sum_{j \in \mathcal{S}_B} \beta_j (\mathbf{y}_r \mathbf{x}_r^\top \mathbf{y}_j \mathbf{x}_j) = \mu \cdot \alpha_r + \sum_d [\mathbf{z}_{\mathcal{S}_B}]_i \cdot [\mathbf{z}_r]_i,
\tag{33}
$$

where $\alpha_r \sim \mathcal{N}(0, k \cdot \sigma^2 (B^v + B_1^2 + B_2^2))$ and $[\mathbf{z}_{\mathcal{S}_B}]_i \sim \mathcal{N}(0, \sigma^2 B^v)$.

$$
\mathbf{a}_r^t = \log T + c_r + \xi_r, \text{ s.t. } \xi_r = \mu \cdot \alpha_r + \sum_d [\mathbf{z}_{\mathcal{S}_B}]_i \cdot [\mathbf{z}_r]_i.
\tag{34}
$$

**Combining both cases**  Recall that $\beta_j > 0$ for all $j$. Therefore, $B^2 = (\sum_{j \in \mathcal{S}_B} \beta_j)^2 > B^v = \sum_{j \in \mathcal{S}_B} \beta_j^2$. Based on the analysis of Soudry et al. [46], we know that $\mathbf{w}_B(t)$ grows as fast as $O(\log t)$. Therefore, both $B_1 = O(\log t)$, and $B = O(\log t)$ (must grow at most as fast as that). From the problem definition, $B_1 \leq \Delta B$, where $\Delta \in [0, 1]$.

Our goal now is to find an epoch $t$ during the second-split training when the rare example is correctly classified with high probability, and the mislabeled example is incorrectly classified with high probability. The maxima is achieved when $\mathbf{a}_r^t \approx -\mathbf{a}_m^t$. We assume that the step sizes are sufficiently small such that we can achieve this condition. Moreover, $c_r, c_m \ll \log T$. Hence, the condition is to find $t$ such that $\xi_r + \log T > 0$ and $\xi_m + \log T < k \cdot \mu^2 \cdot B_1$ with high probability. By symmetry, we must find $\mathbb{P}\left(|\xi_m| < k \cdot \mu^2 \cdot B_1/2\right)$.

$$
\mathbb{P}\left(|\xi_m| > k \cdot \mu^2 \cdot \frac{B_1}{2}\right) \leq \mathbb{P}\left(|\xi_1| > k \cdot \mu^2 \cdot \frac{B_1}{4}\right) + \mathbb{P}\left(|\xi_2| > k \cdot \mu^2 \cdot \frac{B_1}{4}\right),
\tag{35}
$$

where $\xi_1 = \mu \cdot \alpha_m \sim \mathcal{N}(0, \mu^2 \sigma^2 k (B^v + B_1^2 + B_2^2)$ and $\xi_2 = \sum_d [\mathbf{z}_{\mathcal{S}_B}]_i \cdot [\mathbf{z}_r]_i$. For the first term, we can directly use the Gaussian tail bound using Chernoff method.

$$
\begin{aligned}
\mathbb{P}\left(|\xi_1| > k \cdot \mu^2 \cdot \frac{B_1}{4}\right) &\leq 2 \exp \frac{-k^2 \mu^4 B_1^2}{32 \mu^2 \sigma^2 d (B^v + B_1^2 + B_2^2)}, \\
&\leq 2 \exp \frac{-k^2 \mu^4 B_1^2}{32 \mu^2 \sigma^2 d (2B^2)}, \\
&\leq 2 \exp \frac{-k \mu^2 \Delta^2}{c_1 \sigma^2}.
\end{aligned}
\tag{36}
$$

Now, for the second term, using Lemma 3 we have that.

$$
\begin{aligned}
\mathbb{P}\left(|\xi_2| > k \cdot \mu^2 \cdot \frac{B_1}{4}\right) &\leq 2 \exp \frac{-k^2 \mu^4 B_1^2}{c_2 \sigma^4 d B^v} + c_1 \exp(-cd), \\
&\leq 2 \exp \frac{-k^2 \mu^2 \Delta}{c_2 \sigma^2 d} + c_1 \exp(-cd),
\end{aligned}
\tag{37}
$$

since $\frac{\mu}{\sigma} > 1$. Finally, combining both the cases we have that

$$
\mathbb{P}\left(|\xi_m| > k \cdot \mu^2 \cdot \frac{B_1}{2}\right) \leq c_0 \exp \frac{-k^2 \mu^2 \Delta^2}{c \sigma^2 d} + c_1 \exp(-cd),
\tag{38}
$$

This shows that as the dimensionality of the dataset increases, the likelihood of prediction on mislabeled examples being flipped while the rare examples retain their prediction increases exponentially. This concludes the proof of Theorem 4.  □

### A.6 Concentration Inequalities and Additional Lemmas

To make our work self-contained, we supplement the reader with additional Lemmas and Theorems that are helpful tools for proving the theorems in this work. We restate versions of the Hoeffding and Bernstein inequalities as in Chatterji and Long [9].

**Lemma 2** (Soudry et al. [46], Theorem 3). *For any linearly separable dataset $\mathcal{S}_A$ and for all small enough step-sizes $\alpha$, we have*

$$\frac{\mathbf{w}_A}{\|\mathbf{w}_A\|} = \lim_{t \to \infty} \frac{\mathbf{w}^{(t)}}{\|\mathbf{w}^{(t)}\|}.$$

**Theorem 5** (General Hoeffding's Inequality). *Let $\theta_1, \ldots, \theta_m$ be independent mean-zero sub-Gaussian random variables and $a = (a_1, \ldots, a_m) \in \mathbb{R}^m$. Then, for every $t > 0$, we have*

$$\mathbb{P}\left[\left|\sum_{i=1}^m a_i \theta_i\right| \geq t\right] \leq 2 \exp\left(\frac{-c_0\, t^2}{K^2\, \|a\|_2^2}\right),$$

*where $K = \max_i \|\theta_i\|_{\psi_2}$ and $c$ is an absolute constant.*

In our case, since we deal with Gaussian random variables, $\|\theta\|_{\psi_2}$ (sub-Gaussian norm) is same as the variance of the random variable. That is, $\theta \sim \mathcal{N}(0, \sigma^2) \implies \|\theta\|_{\psi_2} = K = \sigma$.

**Theorem 6** (Bernstein Inequality). *For independent mean-zero sub-exponential random variables $\theta_1, \ldots, \theta_m$, for every $t > 0$, we have*

$$\mathbb{P}\left[\left|\sum_{i=1}^m \theta_i\right| \geq t\right] \leq 2 \exp\left(-c_1 \min\left\{\frac{t^2}{\sum_{i=1}^m \|\theta_i\|_{\psi_1}^2}, \frac{t}{\max_i \|\theta_i\|_{\psi_1}}\right\}\right),$$

*where $c_1$ is an absolute constant.*

In our case, let $\mathbf{x}_i \sim \mathcal{N}(0, \sigma^2)$ be independent Gaussian random variables, then for $Z = \sum_i^d X_i^2$,

$$\mathbb{P}\left((|Z - d\sigma^2| \geq t\right) \leq 2 \exp\left(-\frac{c_1}{\sigma^2} \min\left\{\frac{t^2}{d\sigma^2}, t\right\}\right).$$

**Lemma 3** (Gaussian Product). *Let $\mathbf{z}_1 \sim \mathcal{N}(0, \sigma_1^2 \mathbf{I}_d), \mathbf{z}_2 \sim \mathcal{N}(0, \sigma_2^2 \mathbf{I}_d)$ be independent multivariate Gaussian random variables. Then,*

$$\mathbb{P}\left[|\mathbf{z}_1 \cdot \mathbf{z}_2| > t\right] \leq 2 \exp\left(\frac{-c\, t^2}{d\sigma_1^2 \sigma_2^2}\right) + 2 \exp\left(-c_0 d\right)$$

*Proof.* The proof of this lemma is a simplified version of the proof for Lemma 20 in Chatterji and Long [9]. First, consider $\mathbf{z}_2$ to be fixed, and only $\mathbf{z}_1$ to be random. Then from Theorem 5 we have

$$\mathbb{P}\left[\left|\sum_{i=1}^d [\mathbf{z}_2]_i \cdot [\mathbf{z}_1]_i\right| \geq t\right] \leq 2 \exp\left(\frac{-c\, t^2}{\sigma_1^2\, \|\mathbf{z}_2\|_2^2}\right).$$

Also, adapting Theorem 6 such that $Z = \sum_i^d X_i^2 = \|\mathbf{z}_2\|_2^2$, and setting $t = d\sigma^2$

$$\mathbb{P}\left((|\|\mathbf{z}_2\|_2^2 - d\sigma_2^2| \geq t\right) \leq 2 \exp\left(-\frac{c_0}{\sigma^2} \min\left\{\frac{t^2}{d\sigma_2^2}, t\right\}\right),$$

$$\mathbb{P}\left((\|\mathbf{z}_2\|_2^2 \geq 2d\sigma_2^2\right) \leq 2 \exp\left(-c_0 d\right).$$

Coming back to the initial problem and again considering both $\mathbf{z}_1, \mathbf{z}_2$ to be random variables, we have

$$\mathbb{P}\left[|\mathbf{z}_1 \cdot \mathbf{z}_2| \geq t\right] \leq \mathbb{P}\left[|\mathbf{z}_1 \cdot \mathbf{z}_2| \geq t \mid \|\mathbf{z}_2\|_2^2 \leq 2d\sigma_2^2\right] + \mathbb{P}\left[\|\mathbf{z}_2\|_2^2 > 2d\sigma_2^2\right], \tag{39}$$

$$\leq 2 \exp\left(\frac{-c\, t^2}{d\sigma_1^2 \sigma_2^2}\right) + 2 \exp\left(-c_0 d\right). \tag{40}$$

$\square$

**Corollary 1** (Chatterji and Long [9], Lemma 20). *There is a $c \geq 1$ such that, for all large enough $C$, with probability at least $1 - \delta/6$, for all $i \neq j \in [n]$,*

$$|\mathbf{z}_i \cdot \mathbf{z}_j| < c \left( \sigma^2 \sqrt{d \log(n/\delta)} \right).$$

**Lemma 4** (Gaussian Square). *There is a $c \geq 1$ such that, for all large enough $C$, with probability at least $1 - \delta/6$, for all $k \in [n]$,*

$$\frac{d\sigma^2}{2} \leq \|\mathbf{z}_k\|_2^2 \leq \frac{3d\sigma^2}{2}.$$

*Proof.* Recall that $\mathbf{x}_k = \boldsymbol{\mu}_k + \mathbf{z}_k$, where $\mathbf{z}_k \sim \mathcal{N}(0, \sigma^2 \mathbf{I}_d)$. Then,

$$\|\mathbf{z}_k\|_2^2 = \sum_i^d [\mathbf{z}_k]_i^2 = Z$$

Adapting Theorem 6, and setting $t = \lambda d\sigma^2$ with $0 < \lambda < 1$

$$\mathbb{P}\left( |\|\mathbf{z}_2\|_2^2 - d\sigma^2| \geq t \right) \leq 2 \exp\left( -\frac{c_1}{\sigma^2} \min\left\{ \frac{t^2}{d\sigma^2}, t \right\} \right),$$

$$\mathbb{P}\left( |\|\mathbf{z}_2\|_2^2 - d\sigma^2| \geq \lambda d\sigma^2 \right) = 2 \exp\left( -\frac{c_1}{\sigma^2} \min\left\{ \lambda^2 d\sigma^2, \lambda d\sigma^2 \right\} \right),$$

$$= 2 \exp\left( -c_1 \lambda^2 d \right). \qquad \text{(since } 0 < \lambda < 1\text{)}$$

Recall that $d \geq C \log(n/\delta)$. We can set $\lambda = 1/2$ so that, $\|\mathbf{z}_2\|_2^2 > d\sigma^2/2$ with probability at least $1 - \delta/6n$ (for large enough $C$). Taking a union bound over all examples gives us the desired result.

Note that we can get closer to $d\sigma^2$ by chosing an appropriately higher value of $C$.

$\square$

**Lemma 5** (Dataset Separability). *With probability at least $1 - \delta$ over random samples of dataset $\mathcal{S}_A$, samples $(\mathbf{x}_1, \mathbf{y}_1), \ldots, (\mathbf{x}_n, \mathbf{y}_n)$ are linearly separable.*

*Proof.* We will show that there exists a set of weights that with high probability correctly classify each example in the dataset. Let $\mathbf{x}_i = \boldsymbol{\mu}_i + \mathbf{z}_i$ for every data point in $\mathcal{S}$. From assumptions, our dataset $(\mathcal{S}_A)$ contains $\mathbf{z}_r, \mathbf{z}_m$ that belong to rare and mislabeled groups. Rest all points are denoted by $\mathbf{z}_k$. Consider the classifier $\mathbf{w} = \sum_i^n \mathbf{y}_i \mathbf{x}_i$. Then,

**Case 1: Rare Example** $(\mathbf{x}_r, \mathbf{y}_r)$

$$\begin{aligned}
\mathbf{y}_r \mathbf{w} \cdot \mathbf{x}_r &= \sum_j \mathbf{y}_j \mathbf{y}_r \mathbf{x}_j \cdot \mathbf{x}_r, \\
&= \mathbf{x}_r \cdot \mathbf{x}_r + \sum_{i \neq j} \mathbf{y}_r \mathbf{y}_j \mathbf{x}_r \cdot \mathbf{x}_j, \\
&= \boldsymbol{\mu}_r \cdot \boldsymbol{\mu}_r + \mathbf{z}_i \cdot \mathbf{z}_i + \sum_{r \neq j} \mathbf{y}_r \mathbf{y}_j \mathbf{z}_r \cdot \mathbf{z}_j, \qquad \text{(since } \boldsymbol{\mu}_r \cdot \boldsymbol{\mu}_i = 0 \; \forall i \in \{1, 2\}\text{)} \\
&= k\mu^2 + \mathbf{z}_i \cdot \mathbf{z}_i + \sum_{r \neq j} \mathbf{y}_r \mathbf{y}_j \mathbf{z}_r \cdot \mathbf{z}_j, \\
&\geq 0 + d\sigma^2/2 - c_1 n \sqrt{d\sigma^2 \log(n/\delta)} \qquad \text{(using Lemma 4 and Corollary 1)} \\
&> 0
\end{aligned}$$

for $d \geq Cn^2 \log(n/\delta)$.

**Case 2: Mislabeled Example** $(\mathbf{x}_m, \mathbf{y}_m)$   Without loss of generality, assume that the mislabeled example is sampled from the distribution $\mathcal{D}_1$, and the set of all correctly labeled examples in the dataset $\mathcal{S}_A$ from this distribution be $\mathcal{S}_{A,1}$.

$$
\begin{aligned}
\mathbf{y}_m\mathbf{w}\cdot\mathbf{x}_m &= \sum_j \mathbf{y}_j\mathbf{y}_m\mathbf{x}_j\cdot\mathbf{x}_m, \\
&= \mathbf{x}_m\cdot\mathbf{x}_m + \sum_{i\neq j}\mathbf{y}_m\mathbf{y}_j\mathbf{x}_m\cdot\mathbf{x}_j, \\
&= \boldsymbol{\mu}_m\cdot\boldsymbol{\mu}_m + \mathbf{z}_i\cdot\mathbf{z}_i - \sum_{j\in\mathcal{S}_{A,1}}\boldsymbol{\mu}_m\cdot\boldsymbol{\mu}_m + \sum_{m\neq j}\mathbf{y}_m\mathbf{y}_j\mathbf{z}_m\cdot\mathbf{z}_j, \\
&\qquad\qquad\qquad\qquad (\text{since } \boldsymbol{\mu}_m\cdot\boldsymbol{\mu}_1 = 1, \boldsymbol{\mu}_m\cdot\boldsymbol{\mu}_2 = 0, \boldsymbol{\mu}_m\cdot\boldsymbol{\mu}_r = 0) \\
&\geq 0 + d\sigma^2/2 - n_1 k\mu^2 - c_1 n\sqrt{d\log(n/\delta)}, \\
&\qquad\qquad\qquad\qquad (\text{if } n_1 = |\mathcal{S}_{A,1}|, \text{ using Lemma 4 and Corollary 1}) \\
&\geq d\sigma^2/2 - nk\mu^2 - c_1 n\sigma^2\sqrt{d\log(n/\delta)}, \\
&> 0
\end{aligned}
$$

for $d \geq C\max\{n^2\log(n/\delta), nk\mu^2/\sigma^2\}$.

**Case 3: Majority Example** $(\mathbf{x}_i, \mathbf{y}_i)$   The case of majority examples directly follows from the case of mislabeled examples. Rather than having a negative summation over the mean vector for n examples (in line 3), we will have a positive summation because the true label will match the label of the rest of the examples in the subset $\mathcal{S}_{A,1}$. This will make the expected value of the dot product even larger.

*Remark:* Since the first split dataset $\mathcal{S}_A$ is separable, it directly follows that the second split dataset $\mathcal{S}_B$ is also separable since it does not have any mislabeled and rare examples.

$\square$

### A.7   Lemmas for Theorem 4

**Lemma 6** (Sign of $\beta$). $\beta_j \geq 0$ for all j in Equation 25.

*Proof.* Analyzing the steps of gradient descent, we have:

$$
\begin{aligned}
\dot{\mathbf{w}}(t) &= -\nabla\mathcal{L}(\mathbf{w}(t)) \\
&= \sum_{j\in\mathcal{S}_B}\exp\left(-\mathbf{y}_j\mathbf{x}_j^\top\mathbf{w}(t)\right)(\mathbf{y}_j\mathbf{x}_j^\top) \\
\mathbf{w}(t) - \mathbf{w}(0) &= \sum_{j\in\mathcal{S}_B}\left((\mathbf{y}_j\mathbf{x}_j^\top)\underbrace{\int_0^t\exp\left(-\mathbf{y}_j\mathbf{x}_j^\top\mathbf{w}(t)\right)dt}_{\beta_j(t)}\right),
\end{aligned}
\tag{41}
$$

Hence, $\beta_j \geq 0 \ \forall\ j$. $\square$

**Lemma 7** (Support Vectors). *If dataset $\mathcal{S}_A$ is separable, then $(\mathbf{x}_m, \mathbf{y}_m)$ and $(\mathbf{x}_r, \mathbf{y}_r)$ are support vectors for $\mathcal{S}_A$.*

*Proof.* We will prove by contradiction. From our assumption, $(\mathbf{x}_m, \mathbf{y}_m)$ and $(\mathbf{x}_r, \mathbf{y}_r)$ are the only mislabeled and rare examples in the first-split $\mathcal{S}_A$ from their respective sub-groups. Let us assume that they are not support vectors. Then, we can directly follow from the Asymptotic Analysis in Subsection A.4 that the probability of correct classification of the rare example is 0.5 and for the mislabeled example approaches 0 as the model is trained for infinite time. But we know that the model achieves 100% accuracy on the training set $\mathcal{S}_A$ with weights $\mathbf{w}_A(T)$. Hence, this is a contradiction, and $(\mathbf{x}_m, \mathbf{y}_m), (\mathbf{x}_r, \mathbf{y}_r)$ must be support vectors for the original classification problem. $\square$

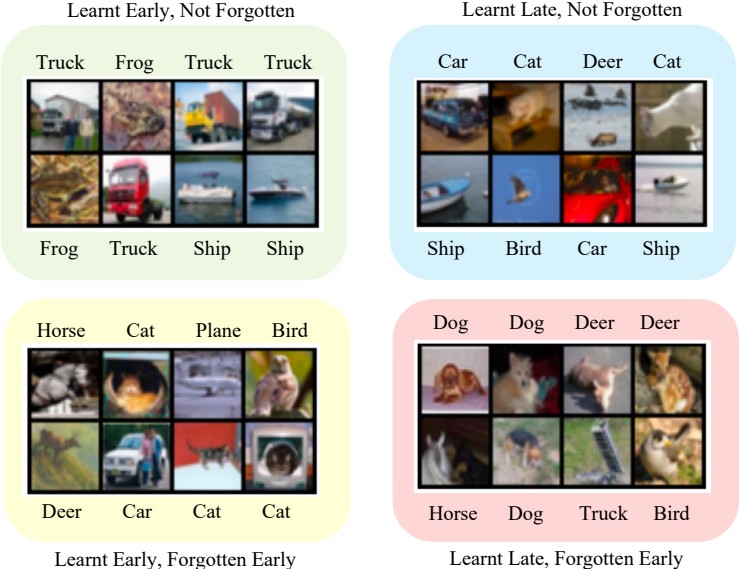

Figure 6: Examples from the CIFAR-10 dataset grouped based on their learning and forgetting time.

**Lemma 8** (Bounded Time Prediction). *With probability at least $1 - \delta$, there exists a bounded time T' = O(T) when the mislabeled examples are incorrectly classified with high probability.*

*Proof.* In Subsection A.4 we have shown that at infinite time training, the model misclassifies mislabeled examples with high probability. Then, the existence of bounded time weights for which the prediction of mislabeled examples is flipped directly follows from the proof of Proposition 1 applied to the result of Theorem 3. $\square$

# B  Experimental Results

## B.1  Experimental Setup

**Architectures**   We perform experiments using four different model architectures—LeNet, ResNet-9 [4], ResNet-50, and Bert-base-cased [14]. Comparisons with model architectures are used in analysis of stability of the SSFT metric. For other numbers reported in tables and plots, we use the ResNet-9 model, unless otherwise stated.

**Optimizer**   We experiment with three different learning rate scheduling strategies—cyclic learning rate schedule, cosine learning rate, and step decay learning rate. We test for two values of peak learning rate—0.1 and 0.01. All the model are trained using the SGD optimizer with weight decay 5e-4 and momentum 0.9, apart from the comparison with optimizers in Appendix C.1 where we also experiment with the Adam optimizer.

**Training Procedure**   We train for a maximum of 100 epochs or until we have 5 epochs of 100% training accuracy. We first train on $\mathcal{S}_A$, and then using the pre-initialized weights from stage 1, train on $\mathcal{S}_B$ with the same learning parameters. All experiments can be performed on a single RTX2080 Ti.

## B.2  Image Datasets

In the main paper we present visualizations of training examples from the MNIST dataset based on which quadrant they lie on in the learning-forgetting graph. Here, we complement our findings by showing visualizations for the CIFAR-10 dataset. We note that CIFAR-10 dataset provide many different types of visibility patterns within the same class. Hence, examples may be learnt late due to belonging to a rare visibility pattern. In Figure 6, we see that the examples that were learnt earliest and

never forgotten have similar visibility patterns—for instance all the trucks have a similar perspective. On the contrary, as we move to the first quadrant with examples that were learnt late but never forgotten, we see that all the examples are true to their semantic class, but these visibility patterns occur rarely. Finally, we also analyze the visualizations based on examples that were forgotten during the course of second-split training. While in the case of MNIST dataset, SSFT was able to remove the mislabeled examples well, we see that CIFAR-10 offers more challenges because examples may be ambiguous because of other reasons and may be forgotten owing to the model using spurious features.

## C   Ablation Studies

We detail the experimental setup used to conduct our ablation studies directed towards understanding the learning and forgetting dynamics of rare and complex examples respectively.

**Rare Examples**   The experiments to show the rate of learning for rare examples are inspired by the singleton hypothesis as proposed by Feldman [15]. The hypothesis was inspired by a long-tailed distribution of visibility patterns in the person and bus category of the PASCAL dataset. For example, the dataset contains many buses with the front visible, but very few buses that were captured from the rear or the side, and even fewer buses whose view is occluded by the presence of other objects infront of them. (Refer to Figure 1 in their work for more details.) In our work, we first attempted to leverage the same training set-up with the provided visibility patters. However, we noted that there wasn't a strong correlation between the frequency of an example's visibility pattern, and the rate at which it was learnt. We hypothesize that this is because there are other factors of example hardness that may make an example be learnt slowly or fast (such as complexity, as detailed in the next paragraph). This can lead to an example being learnt fast if it has a simple pattern yet occurs rarely. Especially when there are only $O(1)$ samples from a given sub-group (based on the visibility pattern), we can not make any claims based on singleton correlation alone.

Hence, in order to distill the frequency of occurrence of an example with other confounders that may influence its training-time, we created a long-tailed dataset from the CIFAR-100 dataset. CIFAR-100 is a dataset of 100 object classes, which can be further grouped into 20 super-classes. For instance, examples from categories *maple, oak, palm, pine, willow* all belong to the 'superclass' of *trees*. Similar division of 5 sub-classes is provided in the datasets for each of the superclasses. Each class contains 500 training examples and the overall dataset has 50,000 training examples.

As a first step towards creating a long-tailed dataset, we assign a fixed frequency ordering within the subgroups of a superclass. The most frequent subgroup has 500 examples in the training set, for the next most frequent subgroup, we randomly select 250 examples from the training set, and so on until the last sub group with 31 examples in the training set. This means that there are exactly 20 sub-groups in the final dataset with {500,250,125,62,31} examples respectively. Irrespective of the class number, the task is to predict the corresponding superclass, that is, we reduce the problem to a 20-class classification problem. However, we track the learning and forgetting dynamics of examples from each of the 100 sub-groups separately, based on their group frequency. To remove any other confounders of example hardness, we (i) randomize the group frequency ordering of the sub-groups within a superclass (in case some classes are harder to learn than the others); and (ii) randomize the examples that were selected based on the group size (in case some examples were ambiguous or hard). We further split the dataset into two IID partitions to analyze the learning time and SSFT, and average the results over 20 random runs of the experiment. Experimental results are detailed in the main paper.

**Complex Examples**   Prior works advocating for, and understanding the simplicity bias [43] have operationalized the notion of simplicity via the complexity of hypothesis class required to learn the distribution that a complex example may be sampled from. In particular, Shah et al. [43] construct a synthetic dataset with MNIST and CIFAR-10 images vertically stacked on top of each other—with the part with MNIST images corresponding to the part of the combined image with *simpler* features, and the part with CIFAR-10 images corresponding to the part with *complex* features. They show that the model almost completely relies on the part of the image containing the MNIST digit even when it is less predictive of the true label. Inspired by this argument about the simplicity of features, we create a dataset that has the the union of images from the MNIST and the CIFAR-10 dataset. More specifically, we select classes from the MNIST dataset corresponding to digits $\{0, 1, 2, 3\}$, and classes from the CIFAR-10 dataset corresponding to {horses, airplanes, dog, frog} and label them

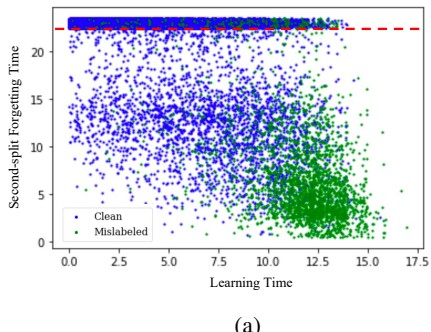 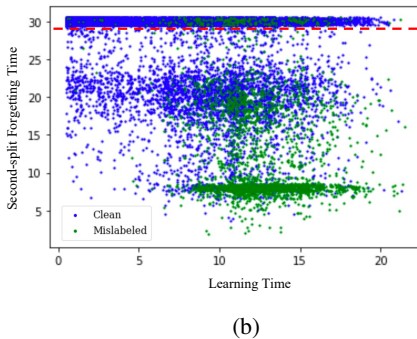

|(a)|(b)|

Figure 7: FSLT (First-split learning time) is able to provide some degree of separation between mislabeled and clean samples when trained with the SGD optimizer (left), but fails when the model is trained using Adam (right) on the CIFAR-10 dataset.

from $\{0, 1, 2, 3\}$. This means that the model associates the label 0 to both the digit 0 and airplane class. The attempt of this experiment is to draw the link between the simplicity bias and the rate of learning. Experimental results are provided in the main paper.

## C.1 Stability of our metric

**Stability across architectures**   The forgetting of examples is a property of both the dataset and the model architecture. As a result, we find that just like the learning time, the forgetting time has a lower correlation between architectures. The average pearson correlation between the ResNet-9 and ResNet-50 models is 0.62 in case of the CIFAR10 dataset. However, we note that the most forgotten examples generalize across datasets. That is, the average pearson correlation between the bottom 10% examples of the dataset is 0.87. This highlights how the forgetting metric is good for finding misaligned examples in the dataset, since they are not a property of the model architecture. We suspect that among the examples that are infrequently forgotten, the model campacity and other inductive biases of the model architecture may have a role in driving the average pearson correlation low.

**Stability across optimizers**   Jiang et al. [25] showed that changing the learning optimizer from SGD to Adam can lead to a significant change in the learning rate of examples from different levels of hardness (based on their regularity metric). More specifically, they find that examples with a low consistency score (closely correlated with learning speed) also get learnt fast when using the Adam optimizer. This suggests that using an optimizer like Adam at training time may have an impact on the ability of learning time based metrics to separate examples. In Figure 7, we contrast the ability of forgetting and learning time based metrics for identifying label noise when using the SGD and Adam optimizers. When using an optimizer such as SGD, the mislabeled samples are learnt slower than a large fraction of the training examples, and the learning time metric offers some degree of separation between the clean and mislabeled examples. However, when we use the Adam optimizer, it results in joint learning of a large fraction of both mislabeled and clean samples. Hence, offering a very low degree of separation. However, under the same training procedure, the SSFT still allows us to distinguish between the mislabeled and clean samples.

**Stability across seeds and learning rates**   The pearson correlation for stability across seeds for the forgetting time metric is 0.83. This is higher than the corresponding learning time based metric (correlation 0.56). However, one of the drawbacks of our proposed metric is that the SSFT requires the use of an appropriate learning rate that allows the examples to be forgotten slowly. We provide more information about the same in the main paper.

**Stability Across Learning Rate Schedules**   We experiment with three different learning rate schedules for the second-split training—triangular, cosine and linear. In triangular learning rate, we increase the learning rate from 0 to the maximum set value linearly over the first 10 epochs, and then decay it back to 0 until we reach the last epoch (maximum of 100 epochs). In case of the linear schedule, we increase the learning rate from 0 to the maximum set value linearly over the course of 100 epochs. The intuition behind using a linear learning rate schedule was to be able to set higher

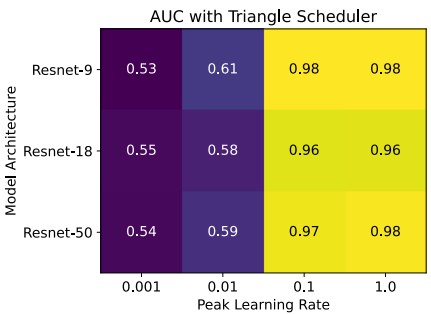

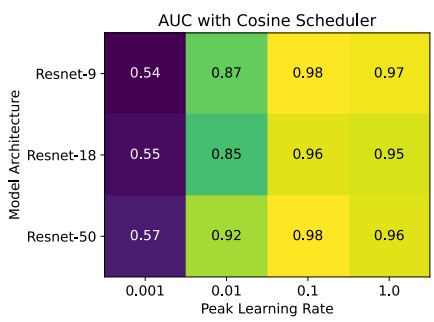

(a) Triangular Learning Schedule          (b) Cosine Learning Schedule

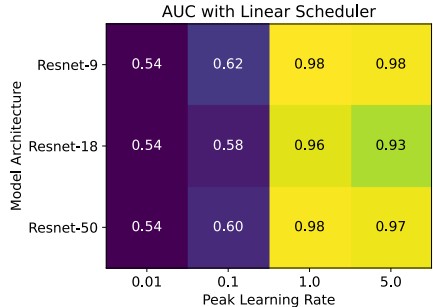

(c) Linear Learning Schedule

Figure 8: We present a heatmap for the AUC of detection of mislabeled examples using the SSFT metric under various learning rates, architecture sizes and learning rate schedules for the SGD optimizer. The experiment was performed on the CIFAR-10 dataset with 10% label noise, and the forgetting times were averaged over 5 seeds before using them for AUC calculation.

peak learning rates so that the model eventually forget all the examples in the first split and we can create a better ordering between samples based on forgetting time (as opposed to the setting where only a small fraction of examples are ever forgotten).

The results of the combined analysis across hyperparameters such as architecture, learning rate and learning rate schedule are presented in the heatmap of AUC of mislabeled example detection in Figure 8. The experiment was performed on the CIFAR-10 dataset with 10% label noise, and the forgetting times were averaged over 5 seeds before using them for AUC calculation. We can see that uniformly across architectures and learning rate schedules having a very low learning rate makes nearly all examples indistinguishable based on forgetting time. This is because all the models are sufficiently overparametrized to memorize all the examples in the dataset. Hence, when using a very small learning rate the optimization step moves the model weights insignificantly and we do not forget many mislabeled samples from the first split. On the other side, having a large learning rate helps achieve a strong separation between mislabeled and clean examples which also shows up in the form of high AUC values in the figure.

### C.2 Impact of Sampling Frequency of Mislabeled Examples

In the synthetic experiment performed in Section 4.2, we assumed that mislabeled examples occur from the majority subgroups. As a result, we observed that they get forgotten quickly during second-split training. However, in this section we aim to understand the impact of sampling frequency on the forgetting time of mislabeled examples. More specifically, we now assume that mislabeled samples occur in rare subgroups in the synthetic setup. We find that the learning curves of the mislabeled example stays the same as before, but the forgetting time for mislabeled examples closely approaches that of rare examples. This is because there is very little signal for the model to learn the opposite class during second split training since the example occurs only O(1) times. The learning and forgetting curves pertaining to the same experiment are presented in Figure 9. In contrast with the forgetting

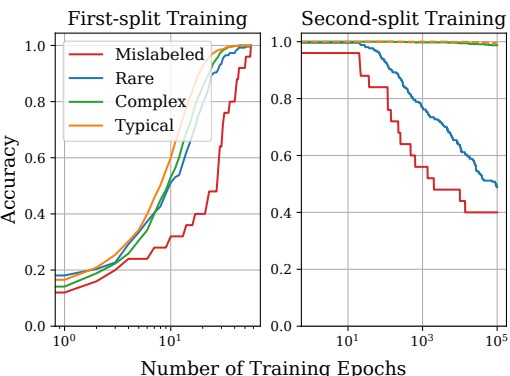

Figure 9: Learning and forgetting curves for mislabeled and rare examples when the mislabeled examples are drawn from rare subgroups in the synthetic setup described in Section 4.2.

curve in Section 4.2 where the mislabeled examples are quickly forgotten and their prediction is flipped, we find that when the subgroup corresponding to the mislabeled examples is infrequent, their forgetting time closely corresponds to that of rare examples; and on aggregate their predictions do not get flipped in the epochs that the model was trained for.

## C.3 Mislabeled Example Detection

In this section we provide additional details about the experimental setting for the results presented in Table 2. In case of the CIFAR-100 dataset, we reduce the learning rate for second-split training by a factor of 10, and use batch size of 128. While all the other training procedures in this paper used a cyclic learning rate, for the case of CIFAR-100, we use warm-up based multi-step decay learning rate schedule.[2] The model used for training was ResNet-18, and 10% label noise was added. The training setting for EMNIST is identical to that of the MNIST dataset. We use the first 10 classes of the dataset to make it a 10-class classification problem.

---

[2]We follow the code in https://github.com/weiaicunzai/pytorch-cifar100