# OpenReview forum: "Characterizing Datapoints via Second-Split Forgetting"
_NeurIPS.cc/2022/Conference — NeurIPS 2022 Accept_

### Official Review · Reviewer_AmTF · 2022-07-07

**Rating:** 8
**Confidence:** 3
**Soundness:** 4 excellent
**Presentation:** 4 excellent
**Contribution:** 3 good

**Summary:**

This paper proposes a new (combination) of metrics to characterize datapoints in dataset, namely the First-Split Learning Time (FSLT) and the Second Split Forgetting Time (SSFT). These measure respectively for all the datapoints in the dataset used during training how fast they are learned, and how quickly they are forgotten, when retrained on a held out portion of the training set. The paper evaluates/investigates the relationship between FSLT and SSFT with respect to mislabeled samples, rare samples, and complex samples. This is done on multiple modalities (various (altered/syntetic) image datasets and a sentiment classification dataset. These metrics can effectively distinguish between these different examples, based on inspecting the effect on one/both metrics.

Moreover, the paper also introduces some theoretical results to support their experiments

**Questions:**

A question I have regarding rare examples: what is the added benefit of using FSLT to find rare examples? Isn't it easy to just see how many datapoints there are per class?

**Limitations:**

-

**Strengths And Weaknesses:**

I particularly like this work, because the metrics introduced are simple (anyone familiar with some Deep Learning will understand them quickly), and are shown to be effective through an extensive set of experiments. In my opinion these types of simple metrics are easy to add to deep learning pipelines and can help investigate model behavior and dataset behavior, and guide the improvement of model training.

Moreover, the paper is extremely well-written in my opinion, with very few spelling mistakes and good formalism and organisation of the paper. The narrative also reads nicely

The only experiment/addition that I could imagine which might be useful is the following: Since overparametrized deep models might not exhibit such nice curves, due to the effect mentioned in lines 228-292) it would be interesting to see if shallower architectures can be used to improve generalization for deeper models. For example, maybe repeat a similar type of experiment as in Figure 3, but where the data that is removed is chosen based on a shallower architecture, and the accuracy is reported on deeper architectures. However, this comment is not highly critical, so I don't encourage the authors to do this experiment during the rebuttal if there is no time.

Additional experiments could possibly be done on larger models to show the discrepancy with smaller models. However I don't consider this a major weakness.

minor comments:
- there is a space in front of the colon on line 109
- on line 164, should $\mathcal{I}_i \cup \mathcal{I}_j = \phi$ also have a condition that $i\neq j$?
- Another suggestion is to increase the font inside the figures slightly, as they are a bit difficult to read.

---

> ### Author Response · Authors · 2022-08-02
> **Suggested Experiments and Fixes**
>
> Thank you for your positive assessment and constructive feedback on our work. We have performed a study on the impact of architecture size and learning rate on the forgetting time of various data points.
>
> **Impact of Architecture Size**: We find that the architecture size and scheduler strategy do not have a significant impact on the AUC for finding mislabeled examples. Our analysis suggests a high correlation between different runs over the same learning rate when architecture or scheduler was varied. In the results in the Tables below, we averaged forgetting times for each example over 5 random runs and then computed the AUC for finding mislabeled examples on CIFAR-10.  A more detailed analysis with different learning rate schedulers is also presented in the comment to R3 and **the corresponding [heatmaps](https://tinyurl.com/heamp) have been added to Appendix** and in the preceding link.
>
> **AUC with Triangle Scheduler**
>
> | Model \ LR    	| 0.001 	| 0.01 	| 0.1  	| 1.0  	|
> | ----------- | ----------- | ----------- | ----------- | ----------- |
> | ResNet-9  	| 0.53  	| 0.61 	| 0.98 	| 0.98 	|
> | ResNet-18 	| 0.55  	| 0.58 	| 0.96 	| 0.96 	|
> | ResNet-50 	| 0.54      	| 0.59     	|  0.97    	|    0.98  	|
>
>
> **SSFT on a shallower architecture, and training on deeper architectures**: This is indeed a great suggestion. We will perform the analysis for the camera-ready version of the paper.
>
> **Minor Comments**: Indeed, i $\neq$ j is an important condition in line 165. We have incorporated this and other minor comments regarding writing and will fix the figures for the camera ready.

---

> > ### Comment · Reviewer_AmTF · 2022-08-05
> > **Thank you for the response**
> >
> > I thank the authors for their response and appreciate the experiments done on larger architectures, which seem to corroborate the findings which were presented in the initial paper submission.
> >
> > I keep my positive initial assessment of the paper and recommend that this paper be accepted.

---

### Official Review · Reviewer_9TS9 · 2022-07-10

**Rating:** 6
**Confidence:** 4
**Soundness:** 3 good
**Presentation:** 4 excellent
**Contribution:** 3 good

**Summary:**

The authors propose a new method for quantifying how "hard" training examples are. Previous works considered methods, such as based on the number of times an example is seen before it is classified correctly or the number of times its predicted label is flipped during training. In this work, the authors propose second-split forgetting time (SSFT), which is the time needed to forget a training example if the model is fine-tuned on a different subset of data. The authors argue that SSFT can distinguish between examples that are hard because the labels are noisy vs. examples that are hard because they come from a rare subpopulation. They also discuss other applications of SSFT. Finally, the authors argue that SSFT is robust to the choice of architecture, optimizer, and so on.

**Questions:**

1- When comparing the improvement on generalization after removing noisy examples, do you train the model on the entire set of data (both sets used in SSFT joined together) or one subset only? Can you please specify exactly how the comparison is done?

2- Can you please generate a figure that shows that SSFT is indeed robust to changes in the learning rate? Please see my comment above.

3- What is $\rho(t)$ in Equation 5?

4- ResNet-9 is a small architecture. I believe ResNet-18 should be used at minimum since it's a common baseline model used in the literature. Is there a reason deeper architectures are not used?

5- I find it hard to understand why "complex examples" are defined to be those that have a high signal-to-noise ratio. I would expect the opposite. Is this a typo? If not, can you please explain the intuition behind this definition?


**Limitations:**

Please see my comments above for the limitations of the work. I don't see any potential negative societal impact.

**Strengths And Weaknesses:**

**Strengths**

1- A new method that can offer insight into datasets and can be useful in identifying and correcting noisy examples.

2- The paper is well-executed and is a pleasure to read.

3- The authors provide precise mathematical definitions in a simplified setting for the concepts they discuss in the paper; e.g. "rare", "noisy", and "complex" examples. However, these are not general definitions; they are specific to the construction they study analytically (i.e. separable examples with a linear decision boundary).

**Weaknesses**

1- The experiments are conducted on a few datasets, even though it should be straightforward to conduct some of the experiments on a larger set of datasets, e.g. for the identification of label noise in Table 2. In particular, the authors used CIFAR100 in other experiments  but without reporting CIFAR100 results in Table 2. Since the claims made in the paper are empirical, the experiments should cover many datasets.

2- The authors claim that SSFT can be used to improve generalization by identifying and removing noisy examples. But, in order for this to be indeed a useful application, the comparison should be against training the model on the *entire* data. In SSFT, the data is split into two subsets and I am not sure if the authors compare the impact on generalization by training on the entire set before and after removing the noisy examples. Please see my question below.

3- As the authors acknowledge in their work, SSFT is sensitive to the choice of the learning rate. The authors claim that for "reasonable" choices, SSFT is robust but there is no evidence of this in the paper. At minimum, the authors can vary the learning rate and report the Pearson correlations (since they used it later to measure stability) in a 2D heat-map with the learning rate on the x and y axis.

4- In some places, the mathematical notation is either imprecise or unclear (to me at least). For example, in Equation 3, $P(x\sim X_g)$ should be $P(x\in X_g)$. In Equation 5, $\rho(t)$ is undefined. Also, when mentioning $O(1)$ in Definition 5.1, this means that the number of rare examples is bounded by a constant *as we vary another parameter*. What is the other parameter? Is it the number of mixtures $N$ or the total number of training examples (in which case $O(1)$ can be a function of $N$)?

5- The experiments are done on a shallow architecture (ResNet-9). I think this is a big weakness of the work. At minimum, ResNet-18 should be used and (naturally) even deeper architectures would be more preferable.

**Minor comments**

- I think the authors should use the "number of seen examples" instead of the number of epochs because the dataset can be quite large (near-infinite data regime) in which a single epoch is used. This is especially important here because the model is *fine-tuned* so one would expect it to converge quickly.

- In line 164, the condition $i\neq j$ should be added to the statement $I_i\cap I_j=\emptyset$.

---

> ### Author Response · Authors · 2022-08-02
> **Requested Experiments and Clarifications**
>
> Thank you for an extremely detailed and constructive review. We have acted upon most of your suggestions which make the results of our work even more compelling. We provide line-by-line clarification for all of your questions below.
>
> 1. **Impact of Learning Rate and Architecture Size**: Based on your suggestion, we performed a detailed analysis of the impact of learning and architecture size across 4 learning rates, 3 architectures, and 3 learning rate schedulers. Across all experiments, we find that the architecture size and scheduler strategy do not have a significant impact on the AUC for finding mislabeled examples.
> Having a very low learning rate allows deep networks to learn examples from the new dataset without perturbing the initial model weights significantly. This is because the stopping condition for model training is the epoch at which we achieve 100% accuracy on the second split.
>     - Hypothetically, if we were to train on the second split for infinite time, we would not expect a low learning rate to have a significant impact.
>     - Our results also suggest a high correlation between different runs over the same learning rate when architecture or scheduler was varied. In the results in the Tables below, we averaged forgetting times for each example over 5 random runs and then computed the AUC for finding mislabeled examples. **The corresponding [heatmaps](https://tinyurl.com/heamp) have been added to Appendix** and in the preceding link.
>
> **AUC - Triangle Scheduler**
>
> | Model \ LR    	| 0.001 	| 0.01 	| 0.1  	| 1.0  	|
> | ----------- | ----------- | ----------- | ----------- | ----------- |
> | ResNet-9  	| 0.53  	| 0.61 	| 0.98 	| 0.98 	|
> | ResNet-18 	| 0.55  	| 0.58 	| 0.96 	| 0.96 	|
> | ResNet-50 	| 0.54      	| 0.59     	|  0.97    	|    0.98  	|
>
> **AUC - Cosine Scheduler**
>
> | Model \ LR     	| 0.001 	| 0.01 	| 0.1  	| 1.0  	|
> | ----------- | ----------- | ----------- | ----------- | ----------- |
> | ResNet-9  	| 0.54  	| 0.87 	| 0.98 	| 0.97 	|
> | ResNet-18 	| 0.55  	| 0.85 	| 0.96 	| 0.95 	|
> | ResNet-50 	| 0.57      	|  0.92    	|  0.98    	| 0.96      	|
>
> **AUC - Linear Scheduler**
>
> | Model \ LR     	| 0.01 	| 0.1 	| 1.0  	| 5.0  	|
> | ----------- | ----------- | ----------- | ----------- | ----------- |
> | ResNet-9  	| 0.54  	| 0.62 	| 0.98 	| 0.98 	|
> | ResNet-18 	| 0.54  	| 0.58 	| 0.96 	| 0.93 	|
> | ResNet-50 	| 0.54      	| 0.60      	| 0.98     	| 0.97     	|
>
>
> 3. **More Datasets**: We will add results for more image datasets via a follow-up comment during the discussion phase.
>
>
> 4. **Comparing generalization accuracies**: For all models, we compare the generalization accuracies if only the first split (half of the dataset) was available for training. We remove examples from the first split based on second-split forgetting time and retrain the model (on the first split) using the exact same procedure. This can be extended in a straightforward way to the whole dataset by computing the forgetting time for all examples in the second-split (this time w.r.t. the first split!). Then we will jointly train on the remaining examples in the union of the first and second split. In fact, we can further increase the SSFT correlation with pathological samples by using multiple random splits of the dataset (not just 2) and averaging out the forgetting time of an example across these splits. This will of course come at the expense of more computation time, but help make the results more robust. We will do this for the camera-ready version (and try to provide updates as we build these models, but this may not be fully complete by the end of the discussion period). However, the key trends are expected to stay unchanged, and the main point of the experiment in this paper was to show a potential use-case of the method and not improve on the state of art generalization.
>
> 5. **Notation Clarity and Preciseness**: Thank you for pointing out these issues. We have updated all the notational issues in the paper and added descriptions wherever required.
>    - $\rho(t)$ is the residual term that is bounded (does not grow as log(T)). We have now mentioned this in the paper.
>    - Indeed, i $\neq$ j is an important condition in line 165. We have incorporated your minor comments regarding writing.
>    - O(1) is w.r.t. the training set size.
>
> 6. **Complex Examples**: This was a typo! thank you for pointing it out. Indeed, complex examples are modeled as those with high noise and low signal, hence requiring a higher example complexity to learn.
>
> 7. **Number of Seen Examples**: The number of batch iterations would indeed be a more useful metric than the number of epochs. We will update the graphs to reflect this in the main paper. However, evaluating the accuracy of each example in the training set after each iteration will be extremely expensive. Hence, we will still evaluate after a fixed number of iterations (typically the same as the number of iterations in an epoch for 50,000-sized datasets).

---

> > ### Comment · Reviewer_9TS9 · 2022-08-04
> > **Follow up**
> >
> > Thank you. I think this addresses most of my comments. But, I am still unclear about the practical application.
> >
> > To elaborate on my point, suppose that I have a training dataset of size $n$. I have the option of training the model on all of the $n$ examples, including the noisy ones. This is the baseline method.
> >
> > The second option is to use SSFT. But, to do that, I would train on $n/2$ examples and fine-tune the model on the second subset to identify noisy examples in the first half. Suppose that there are $m$ noisy examples identified by SSFT. I can then remove them and train the model on the remaining $n-m$ examples. Does this perform better than training on all $n$ examples?
> >
> > In your evaluation, what you do is different if I understand it correctly. You compare training on the first $n/2$ examples with training on the $n/2-m$ examples (all from the first half of the dataset). But, in practice, the default approach is not really to train on $n/2$ examples; it's to train on all $n$ examples.
> >
> > Can you please clarify which one you use? This needs to be made more clear in the paper.

---

> > > ### Author Response · Authors · 2022-08-05
> > > **Clarification**
> > >
> > > Thank you for following up with your concern. We will attempt at restating the method as proposed in point 4 in our first response:
> > >
> > > 1. Divide dataset $D$ into $D_1, D_2$ of size $n/2$ each. The goal is to remove $m$ examples (e.g. $m$ noisy examples from the dataset $D$).
> > > 2. Train on $D_1$ to $100\%$ training accuracy. Now, note the forgetting time of each example in $D_1$ by fine-tuning on $D_2$.
> > > 3. Using a new randomly initialized model, repeat step 2 by swapping $D_1$ and $D_2$. That is, train on $D_2$ to 100% accuracy. Now, note the forgetting time of each example in $D_2$ by fine-tuning on $D_1$.
> > > 4. Now we have the forgetting time of each example in $D$. Remove the top m examples in D.
> > > 5. Train on the remaining n-m examples.
> > >
> > > The above is a natural way in which the split training can be extended to calculate forgetting times for each example.
> > >
> > > **In your evaluation, what you do is different … it's to train on all n examples.**
> > > Yes, indeed, for our current evaluation, we use only the first split for training, but the trends in the graphs should stay unchanged even when we follow the aforementioned proposal.

---

> > > > ### Comment · Reviewer_9TS9 · 2022-08-07
> > > > **Response**
> > > >
> > > > Yes, it seems reasonable that the same results would hold if you follow this approach. Thanks for the clarification.

---

> > > > > ### Author Response · Authors · 2022-08-09
> > > > > **Additional Results for AUC calculation**
> > > > >
> > > > > As per your suggestion, we have also run the AUC calculation on two more vision datasets, CIFAR100 and EMNIST. The combined table of results is presented below (we will merge the whole table in the main paper using the additional space provided in the final version). We find that in the case of EMNIST all the methods have very high AUC for detecting mislabeled examples since the dataset is much simpler than CIFAR10 and CIFAR100. In the case of CIFAR100, we find that **ssft** performs significantly better than its counterpart **fslt**. The first split metric can be significantly improved by aggregating cumulative accuracies across all examples. Finally, using the joint method of both the first and second-split ranks provides the best AUC for mislabeled example detection. We will provide more details about the training procedure in Appendix C.3.
> > > > >
> > > > > **Method $\to$**|**$\mathbf{lsft}$**|**$\mathbf{acc}\_l$**|**$\mathbf{ssft}$ (Ours)**|**$\mathbf{acc}\_f$ (Ours)**|**$\mathbf{conf}\_l$**|**$\mathbf{n}\_f$**|**Joint (Ours)**
> > > > > :-----:|:-----:|:-----:|:-----:|:-----:|:-----:|:-----:|:-----:
> > > > > Imagenette|0.834|0.912|0.931|0.941|0.786|0.781|0.957
> > > > > CIFAR10|0.740|0.900|0.938|0.941|0.947|0.580|0.958
> > > > > MNIST|0.973|0.998|0.997|0.998|0.965|0.377|0.998
> > > > > CIFAR100|0.700|0.899|0.865|0.885|0.860|0.300|0.926
> > > > > EMNIST|0.987|0.990|0.987|0.989|0.984|0.386|0.997
> > > > >
> > > > > We hope we were now able to address all your queries and look forward to clarifying any remaining concerns. Thank you for your detailed feedback and engagement throughout the review cycle!

---

### Official Review · Reviewer_nsE6 · 2022-07-11

**Rating:** 8
**Confidence:** 5
**Soundness:** 3 good
**Presentation:** 4 excellent
**Contribution:** 4 excellent

**Summary:**

The paper proposes a new approach to analyse the learning and forgetting of examples in training deep neural networks, working on range of domains from vision to language. This approach shows promise in identifying *hard* examples, and differentiating rare classes mislabelled examples in particular.

**Questions:**

- It seems quite natural that neurons firing for classifying a rare class would not be overwritten, given that they are rarely seen. Whereas mislabelled examples (likely more common occurring classes) would overwrite. So the training curves as they are seem intuitively correct. Any comment on this statement? Is there something more complex going on?
- Seems like it would be useful to have a number of synthetic examples to test the above? Say make classes (set a few to be mislabelled only) that are mislabelled as rare as the rare classes.

**Limitations:**

The authors have addressed the limitations of their approach, however not in the relation to societal impact. This does not seem necessary here. Although perhaps a statement on how this approach might impact minority groups (classes).

**Strengths And Weaknesses:**

The paper is a strong paper, and I am quite fond of it. It has implications for the communities which makes specific use of hard examples, noisy labels and rare classes, and more broadly the community interested in learning dynamics. I do have a number of question and concerns which I've listed in the questions section.

Strengths
- Approach is simple yet effective.
- Well written paper.
- Experiments on a large array of tasks and domains.
- Strong ablations to understand how, why and where this approach works.
- Clear results.
- Sections 4.5 and 4.6 are appreciated; drawing attention to utility of the method(s) and failure modes. This will be useful for practitioners using this. Limitations of the approach are clearly described in failure modes.


Weaknesses
- The approach requires a separate hold-out set to train on to describe the dynamics.
- Retraining is computation and time consuming.
- The description of the synthetic dataset in 4.2 could be more readable, clear and precise.


Originality
Novel approach to classifying examples.

Quality:
Theoretically grounded paper, with good intuitions and strong results.

Clarity:
Well written paper. It provides a good section of related work, ablations and explanations on method. Figures were clear and useful.

Significance:
Paper builds on the work of the likes of Toneva et al. classifying hard examples, and has implications for communities interested in e.g. curriculum learning.

---

> ### Author Response · Authors · 2022-08-02
> **Clarifications and Suggested Experiment**
>
> Thank you for your positive feedback on our work. We provide line-by-line clarification for all of your questions below.
>
>
> **Overwriting of neurons**: As you point out in your question, indeed the insight for SSFT stems from the intuition that a model should not overwrite neurons that are supposed to fire only for rare examples, and quickly forget the mislabeled examples because they were memorized purely based on noise and there are other neurons in the second split that fire in the direction of the true label.
>
> 1. *Why should a rare example be forgotten at all?* This can be understood intuitively through the idea of weight decay in deep learning models. A model is incentivized to reduce the norm of the weights and hence overwrite redundant neurons. However, such a phenomenon is expected to occur even in the absence of weight decay (albeit much more slowly) as shown in prior works pertaining to the biases of stochastic gradient descent. This leads the optimization procedure towards the min norm solution at convergence.
>
> 2. *Is there something more complex going on?* We believe you have captured the essence of the phenomenon correctly. The simplicity of the phenomenon is one of the key strengths of SSFT as you also appreciated in your review!
>
> **Suggested Experiment**
> If we understand your comment correctly, the suggestion is to include mislabeled examples in rare sub-classes and see if they are still forgotten. While creating such a setup with no other confounding factors is hard for real datasets (recall that even in our CIFAR100 long-tailed setup we randomize multiple times to remove other confounders), we do perform the suggested experiment in our synthetic setup.  As you correctly guessed, the learning curves for the mislabeled examples indeed follow the rare example learning curve much more closely when this happens during the training time.
>
> We find that the learning curves of the mislabeled example stays the same as before, but the forgetting time for mislabeled examples closely approaches that of rare examples. This is because there is very little signal for the model to learn the opposite class during second split training since the example occurs only O(1) times. We have added this experiment in Appendix C.2 and the figure can also be viewed in the link [here](https://i.ibb.co/51LRwkh/sim-2.png).

---

> > ### Comment · Reviewer_nsE6 · 2022-08-08
> > **Good clarifications**
> >
> > Thanks for the clarifications and additional experiments. I think the paper would be well-served by adding a mention of the limitation for the mislabelled/rare combination in the main manuscript.
> >
> > In addition, good work in response to the other reviewers concerns.
> >
> > I've raised my score to 8.

---

> > > ### Author Response · Authors · 2022-08-09
> > > **Thank you**
> > >
> > > Thank you for your encouraging feedback! We will mention this in the final version of the main paper.

---

### Official Review · Reviewer_bwAK · 2022-07-18

**Rating:** 7
**Confidence:** 2
**Soundness:** 4 excellent
**Presentation:** 4 excellent
**Contribution:** 4 excellent

**Summary:**

This paper studies the training time dynamics and the hardness of examples. It proposes a new metric to complement existing metrics called second-split forgetting time (SSFT). The paper studies three types of hard examples, including mislabeled examples, rare examples, and complex examples. The paper empirically shows that SSFT and FSLT together can be effective in identifying noisy labels, improving generalization in noisy data settings, identifying failure modes, and being robust to random seeds. In addition, the paper studies SSFT in theory based on a toy example.

**Questions:**

This paper proposes a new and interesting metric and the empirical evaluation is comprehensive. Overall, I think this is a solid paper.

**Limitations:**

The authors have adequately addressed the limitations and potential negative societal impact of their work

**Strengths And Weaknesses:**

Strengths:

1. A new metric for characterizing the example hardness.
2. Studied three types of hard examples and verify the effectiveness of the new metric with comprehensive experimental studies.
3. Well-written paper.

Weakness:

1. The theoretical results are derived from a simplified setting.

---

> ### Author Response · Authors · 2022-08-02
> **Thank you for your positive assessment**
>
> Thank you for your positive assessment of our work. Based on common reviewer suggestions, we have highlighted all changes made to improve the paper in the general response.

---

### Author Response · Authors · 2022-08-02
**General Response to Reviewers**

We thank all the reviewers for their detailed feedback and comments. We are pleased to see that all the reviewers are positive about their assessment of our work. Especially,

1. All the reviewers appreciated the **clarity, organization, and quality of the writing** which made the paper easy to follow (R1, R2, R3, R4).
2. Further, multiple reviewers found the **experimental study** to be **comprehensive** with **clean ablation** studies (R1, R2, R4).
3. Reviewers especially appreciated the **simplicity** (R1, R2, R4) and  **novelty** (R1, R2, R3) of our approach which makes it easy to apply to any deep learning work.

Acting upon some of the suggestions made in the reviews has helped further improve the quality of our work. Particularly,
1. Ablation experiments on the impact of learning rate and architecture size on SSFT have been included in Appendix C.1.
2. Experiments are included to show the effect of reduced sampling frequency on mislabeled examples in a synthetic dataset C.2.
3. Experimental results for SSFT on more vision datasets will be included in Appendix C.3 (in an upcoming revision).
4. The assumptions of the theoretical model have been updated slightly, to fit with the previously used setup of Chatterji and Long 2021.
5. We have updated the description of the datasets and cleaned up all minor notation issues.

---

### Meta-Review · Area_Chair_XHFM · 2022-08-22

**Recommendation:** Accept
**Confidence:** Certain

**Metareview:**

The paper studies metric to characterize the difficulty of examples in terms of training dynamics: in addition to first-split learning time (FSLT), similar in principle to existing methods, they propose the novel second-split forgetting time (SSFT). There is strong empirical evidence that SSFT allows to discriminate mislabelled and rare examples, and is a robust metric. The paper also proposes a theoretical analysis in a simplified setting.

The reviewers were unanimous: the method is very well written and provides good intuition. FSLT looks simple, and is yet a very efficient method. A very nice job was done by the authors to characterize various informal notions of difficulty ("rare", "mislabeled" and "complex"). A few minor problems were discussed with the reviewers, and carefully addressed by the authors. We therefore all enthusiastically recommend to accept the paper. Some reviewers spontaneously proposed to highlight the paper (oral or spotlight presentation or award) and I support this.

**Award:**

Yes

---

### Decision · Program_Chairs · 2022-09-14

Accept